# Production of seedable Amyloid-β peptides in model of prion diseases upon PrP$^{Sc}$-induced PDK1 overactivation

Juliette Ezpeleta[1,2,6], Vincent Baudouin[1,2,6], Zaira E. Arellano-Anaya [1,2], François Boudet-Devaud[1,2], Mathéa Pietri[1,2], Anne Baudry[1,2], Anne-Marie Haeberlé[3], Yannick Bailly[3], Odile Kellermann[1,2], Jean-Marie Launay[4,5] & Benoit Schneider[1,2]

The presence of amyloid beta (Aβ) plaques in the brain of some individuals with Creutzfeldt-Jakob or Gertsmann-Straussler-Scheinker diseases suggests that pathogenic prions (PrP$^{Sc}$) would have stimulated the production and deposition of Aβ peptides. We here show in prion-infected neurons and mice that deregulation of the PDK1-TACE α-secretase pathway reduces the Amyloid Precursor Protein (APP) α-cleavage in favor of APP β-processing, leading to Aβ40/42 accumulation. Aβ predominates as monomers, but is also found as trimers and tetramers. Prion-induced Aβ peptides do not affect prion replication and infectivity, but display seedable properties as they can deposit in the mouse brain only when seeds of Aβ trimers are co-transmitted with PrP$^{Sc}$. Importantly, brain Aβ deposition accelerates death of prion-infected mice. Our data stress that PrP$^{Sc}$, through deregulation of the PDK1-TACE-APP pathway, provokes the accumulation of Aβ, a prerequisite for the onset of an Aβ seeds-induced Aβ pathology within a prion-infectious context.

[1] Université Paris Descartes, Sorbonne Paris Cité, UFR des Sciences Fondamentales et Biomédicales, UMR 1124, 75006 Paris, France. [2] INSERM, UMR 1124, 75006 Paris, France. [3] Trafic Membranaire dans les Cellules du Système Nerveux, Institut des Neurosciences Cellulaires et Intégratives, CNRS UPR 3212, 67000 Strasbourg, France. [4] Assistance Publique des Hôpitaux de Paris, INSERM UMR 942, Hôpital Lariboisière, 75010 Paris, France. [5] Pharma Research Department, Hoffmann La Roche Ltd, 4070 Basel, Switzerland. [6] These authors contributed equally: Juliette Ezpeleta, Vincent Baudouin. Correspondence and requests for materials should be addressed to J.-M.L. (email: jean-marie.launay@inserm.fr) or to B.S. (email: benoit.schneider@parisdescartes.fr)

Prion diseases, such as Creutzfeldt-Jakob (CJDs) and Gertsmann-Straussler-Scheinker (GSS) diseases in humans, are neurodegenerative disorders with sporadic, genetic or iatrogenic origins. These pathologies are characterized by the accumulation in the central nervous system of an abnormally folded and toxic protein, the scrapie protein—PrP$^{Sc}$, the trans-conformational isoform of the cellular prion protein PrP$^C$ (ref. [1]). Beyond PrP$^{Sc}$-induced neuropathological lesions, the occurrence of Alzheimer-like pathology was reported in the brain of elderly CJD or GSS patients[2–7] as well as young individual with iatrogenic CJD (iCJD)[8]. Colocalization of β-amyloids (Aβ) with PrP$^{Sc}$ in the same amyloid plaques[9] suggests potential interrelationships between prion pathogenesis and production and/or deposition of Aβ. Accordingly, prion infection with the 139 A scrapie strain of transgenic mice with Alzheimer-like pathology (Tg2576) enhances the cortical accumulation of fibrillar and aggregative Aβ42 (ref. [10]). Clinical examination of the brain of iCJD patients contaminated with PrP$^{Sc}$-containing human growth hormone (hGH) or grafted with prion-infected dura mater showed Aβ deposits in the brain parenchyma as well as a cerebral amyloid angiopathy (CAA)[11–13]. However, the mechanisms whereby prion infection would connect to the rise and deposition of Aβ in prion diseases remain elusive. The accumulation of toxic Aβ peptides concomitantly with PrP$^{Sc}$ also asks the question about the role exerted by Aβ on prion pathogenesis.

Normal PrP$^C$ has been involved in the control of cellular Aβ production. PrP$^C$ is a GlycosylPhosphatidylInositol(GPI)-anchored protein tethered to the outer leaflet of the plasma membrane, where its behaves as a signaling receptor or co-receptor[14–16] or as a scaffolding protein regulating the assembly and functionality of diverse signaling membrane effectors[17]. Although controversial[18], PrP$^C$ may limit the production of Aβ by inhibiting β-secretase (BACE1)-mediated cleavage of the amyloid precursor protein (APP)[19]. PrP$^{null}$-mice display higher levels of brain Aβ than their PrP$^C$ expressing counterparts and productions of Aβ40 and Aβ42 by neuroblastoma cells are cancelled upon PrP$^C$ overexpression[19]. Alternatively, PrP$^C$-dependent activation of the α-secretase TACE (ADAM17)[20] may limit Aβ production. Physiologically TACE contributes to the regulated cleavage of APP into the protective soluble fragment sAPPα, which precludes the production of Aβ peptides by the β-secretase pathway[21,22].

We previously evidenced that deregulation of the PrP$^C$-TACE α-secretase signaling pathway plays a critical role in prion neuropathogenesis[23,24]: PrP$^{Sc}$ induces (i) an overactivation of the 3-phosphoinositide-dependent kinase 1 (PDK1), and (ii) a downstream internalization of TACE. This internalization decreases by more than 80% the TACE shedding activity towards PrP$^C$, then amplifying PrP$^C$ conversion into PrP$^{Sc}$ (ref. [23]). Whether PrP$^{Sc}$ would also reduce APP α-cleavage by TACE, thus favoring the β-amyloidogenic pathway through corruption of the PrP$^C$-PDK1-TACE pathway, has never been investigated.

Combining in vitro and in vivo approaches, we here show that PrP$^{Sc}$-induced overactivation of PDK1 and downregulation of TACE activity shifts APP towards its β-processing and Aβ40/42 overproduction. Aβ40 and Aβ42 accumulate mainly as monomers, but trimers and tetramers of Aβ40/42 are also produced within a prion-infectious context. To address the role exerted by the overproduced Aβ on prion pathogenesis, we built an Aβ-free prion-infected cell system by chronically silencing the expression of APP in 1C11 neuronal stem cells[25] (referred to as APP$^{null}$-1C11 cells) prior to prion infection. The absence of Aβ does not alter prion replication in prion-infected APP$^{null}$-1C11 cells, and cell-based PrP$^{Sc}$ inocula Aβ-free or not display similar prion infectivities when injected to C57Bl/6 J mice. We further show that PrP$^{Sc}$ promotes a similar rise of Aβ monomers in the CSF of

C57Bl/6 J mice whatever the presence of Aβ in PrP$^{Sc}$ inocula. Prion infection of C57Bl/6 J mice also triggers the formation of Aβ trimers and tetramers, but the relative proportion of those Aβ oligomers varies according to the presence or not of Aβ species in PrP$^{Sc}$ inocula. With the help of transgenic APP23 mice, we also provide evidence for brain deposition of PrP$^{Sc}$-induced Aβ, only when seeds of Aβ trimers are co-transmitted with PrP$^{Sc}$, whereas Aβ-free PrP$^{Sc}$ inocula fail to induce any Aβ deposition. Importantly, prion infection combined to brain Aβ deposition reduces the survival time of prion-infected APP23 mice without modifying prion replication, indicating the contribution of amyloid deposition to the death of prion-infected mice.

## Results

**PrP$^{Sc}$ corruption of the PDK1-TACE pathway triggers Aβ rise.**
Exploiting the 1C11 neuronal stem cell line[25] that supports prion replication[26], we measured through liquid chromatography-tandem mass spectrometry (LC-MS/MS) analyses accumulation of soluble Aβ40 and Aβ42 peptides in the surrounding milieu of 1C11-derived serotonergic neuronal cells chronically infected by the Fukuoka-1 prion strain (Fk-1C11$^{5-HT}$ cells) (Fig. 1a–b). We then assessed whether this rise of Aβ in the culture medium of Fk-1C11$^{5-HT}$ cells would relate to a PDK1 overactivity, downstream TACE internalization[23] and subsequent reduction of APP α-cleavage. Pharmacological inhibition of PDK1 by BX912 (1 μM, 1 h) counteracted the prion-induced increase in Aβ40, Aβ42, and sAPPβ levels in the culture medium of Fk-1C11$^{5-HT}$ cells and favored an enhanced production of the neuroprotective sAPPα fragment (Fig. 1a–d). The rescue of APP α-processing in prion-infected neuronal cells upon PDK1 inhibition depended on the restoration of TACE activity, since TACE inhibition with TAPI-2 (100 μM, 1 h) abrogated the decrease of Aβ40/42 and sAPPβ and the reciprocal increase of sAPPα induced by BX912 (Fig. 1a–d). Our results thus indicate that accumulation of Aβ peptides in prion-infected neuronal cells depends on PDK1 overactivity. Further supporting the view that prion infection engages APP towards the β-secretase pathway, we showed that the pharmacological inhibition of BACE1 by MK-8931 (1 μM, 1 h) reduced the Aβ40, Aβ42, and sAPPβ levels in the culture medium of Fk-1C11$^{5-HT}$ cells with an opposite increase in the level of sAPPα (Supplementary Fig. 1). Despite a strong reduction (more than 80%) of plasma membrane TACE level in prion-infected neurons[23], the increase in sAPPα level that typically accompanies BACE1 inhibition likely results from augmented bioavailability of APP for residual α-secretase activities[27].

We next probed the oligomeric state of the Aβ peptides produced by prion-infected 1C11$^{5-HT}$ cells through ion mobility coupled to electrospray ionization mass spectrometry (ESI-IM-MS)[28]. Only Aβ40 and Aβ42 monomers were present in the cell culture medium and lysates of uninfected 1C11$^{5-HT}$ cells whereas in the cell culture medium of Fk-1C11$^{5-HT}$ cells Aβ trimers representing ~2.5% of total Aβ (26.45 ± 2.35 pg ml$^{-1}$ of Aβ40 trimers—i.e., ~2.3% of total Aβ40, and 3.50 ± 0.45 pg ml$^{-1}$ of Aβ42 trimers—i.e., ~3.5% of total Aβ42) were measured. Aβ tetramers representing ~0.25% of total Aβ were also detected (2.35 ± 0.55 pg ml$^{-1}$ of Aβ40 tetramers—i.e., ~0.2% of total Aβ40, and 0.55 ± 0.16 pg ml$^{-1}$ of Aβ42 tetramers—i.e., ~0.5% of total Aβ42, close to the limit of quantification (0.35 pg ml$^{-1}$) of the ESI-IM-MS method used) (Fig. 1e–f). No other forms of soluble Aβ assemblies were evidenced in the culture medium of prion-infected cells, keeping in mind the lower sensitivity of the ESI-IM-MS approach to detect Aβ multimers with oligomeric orders higher than the tetramer (0.20–0.29 pg ml$^{-1}$ for orders 5–8 vs. 0.09–0.16 pg ml$^{-1}$ for orders 1–4, see Methods). In Fk-1C11$^{5-HT}$ cell lysates, the vast majority of Aβ40 and Aβ42

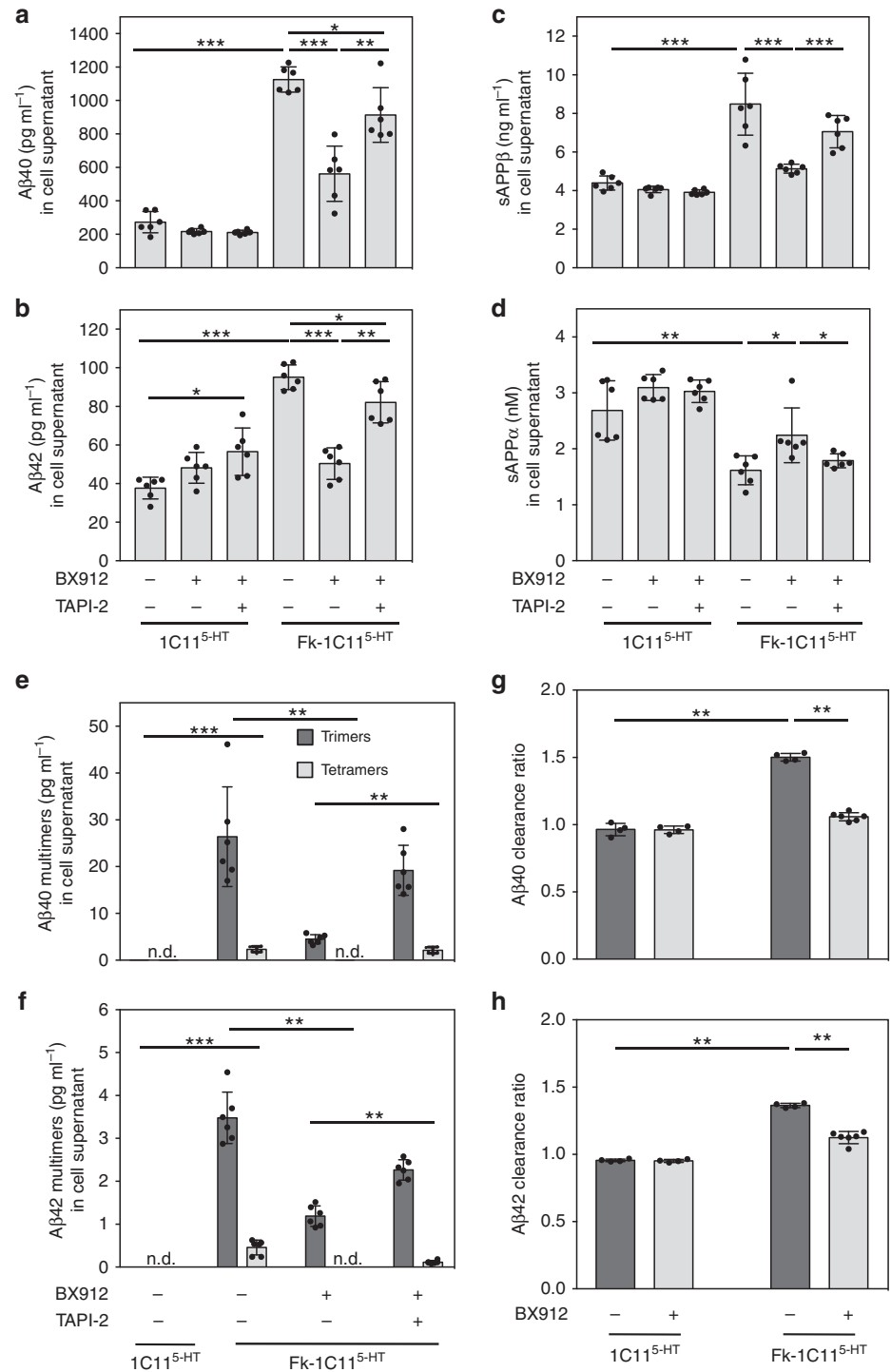

**Fig. 1** PrP$^{Sc}$ deregulation of the PDK1-TACE pathway promotes the accumulation of Aβ monomers and multimers. Concentrations of Aβ40 (**a**), Aβ42 (**b**), sAPPβ (**c**), sAPPα (**d**), and multimers of Aβ40 (**e**) and Aβ42 (**f**) in the culture medium of uninfected serotonergic 1C11$^{5-HT}$ and Fk-infected 1C11$^{5-HT}$ neuronal cells (Fk-1C11$^{5-HT}$) treated or not with the PDK1 inhibitor BX912 (1 μM) or a combination of BX912 (1 μM) and the TACE inhibitor TAPI-2 (100 μM) for 1 h, deduced from LC-MS/MS (**a–d**) and ESI-IM-MS (**e–f**) analyses. **g–h** Aβ40 and Aβ42 clearance ratios measured in the culture medium of 1C11$^{5-HT}$ and Fk-1C11$^{5-HT}$ neuronal cells treated or not with BX912 (1 μM) for 1 h. Values are means ± sem of six independent experiments. n.d. not detected. Data in Fig. 1a–d were analyzed using the two way ANOVA test with Bonferroni post-test correction. Data in Fig. 1g–h were analyzed using the two-tail Student t-test. *denotes $p < 0.05$, **$p < 0.01$, and ***$p < 0.001$. Source data are provided as a Source Data file

displayed a monomeric structure with Aβ40 and Aβ42 trimers and tetramers representing less than 0.20% (42 ± 27 pg mg$^{-1}$ protein) over total Aβ species (24,405 ± 3800 pg mg$^{-1}$ protein). In Fk-1C11$^{5-HT}$ cells BX912-mediated PDK1 inhibition, that lowered Aβ production (Fig. 1a–b), promoted a half to two-third

reduction of Aβ40/42 trimers and rendered tetramers no longer detectable in the culture medium (Fig. 1e–f). TACE inhibition by TAPI-2 counteracted the effect of BX912 on Aβ production and thereby the decrease of Aβ40/42 trimers and tetramers (Fig. 1e–f).

Because an increase of Aβ40/42 in the surrounding milieu of prion-infected neuronal cells might also mirror some dysregulation of Aβ degradation, we measured the Aβ40/42 clearance ratio (i.e., the production over degradation rates of both Aβ40 and Aβ42 peptides) in Fk-infected 1C11[5-HT] cells. To this purpose, prion-infected neuronal cells were incubated with $^{13}C_6$-Leucine for 6 h and the cell supernatants were collected over a 12 h time period. Aβ40 and Aβ42 were quantified through stable-isotope labeling tandem mass spectrometry[29,30]. We found that the Aβ40 and Aβ42 clearance ratios were increased by ~1.5-fold in Fk-infected cells vs. uninfected ones (Fig. 1g–h), accounting for the accumulation of Aβ peptides in the culture medium of prion-infected cells (Fig. 1a–b). Exposure of Fk-1C11[5-HT] neuronal cells to the PDK1 inhibitor BX912 (1 μM, 6 h) significantly reduced the impact of prion infection on the Aβ40 and Aβ42 clearance ratios (Fig. 1g–h).

These in vitro data thus show that PDK1 overactivation induced by prion infection provokes a deficit of the TACE APP α-processing and engages APP towards the β-secretase pathway, favoring the accumulation of Aβ40/42 monomers and multimers.

## Similar infectivity of Aβ-free or Aβ-containing PrP$^{Sc}$ inocula.

We next wondered whether Aβ produced by prion-infected cells would influence prion infectivity. To address this issue, we built an Aβ-free cell system infected by prions exploiting 1C11 neuronal stem cells. Uninfected 1C11 cells were first silenced for the expression of APP using a stable siRNA-based strategy[31]. 1C11 cells stably repressed for APP expression (referred to as APP$^{null}$-1C11 cells) were then chronically infected by the Fukuoka-1 prion strain (referred to as APP$^{null}$-Fk-1C11 cells). We checked in APP$^{null}$-Fk-1C11 cells that Aβ40 and Aβ42 monomers, trimers, and tetramers were no longer detectable in both the culture medium and cell lysates as compared to Fk-1C11 cells by LC-MS/MS (Fig. 2a) and ESI-IM-MS (Fig. 2b) analyses.

Importantly, ELISA-based quantifications of proteinase K-resistant PrP$^{Sc}$ did not significantly differ between Fk-1C11 and APP$^{null}$-Fk-1C11 cells (Fig. 2c).

We next injected to C57Bl/6 J mice via the intracerebral route cell-based inocula either uninfected ($1 \times 10^5$ 1C11 cells) or PrP$^{Sc}$-infected Aβ-free ($1 \times 10^5$ APP$^{null}$-Fk-1C11 cells) or containing Aβ ($1 \times 10^5$ Fk-1C11 cells). Mice inoculated with control 1C11 cells (SHAM) remained healthy over more than 250 days post inoculation, while mice inoculated with Fk-1C11 or APP$^{null}$-1C11 cells both died at comparable times (168.1 ± 3.2 days and 171.8 ± 3.5 days, respectively, $n = 10$, Fig. 3a). Moreover, in mice injected with PrP$^{Sc}$ inocula Aβ-free or not, the mean static rod score dropped below 10 at 160 dpi, indicating similar impairments in motor coordination (Fig. 3b). Accordingly, at the end stage of prion disease highly comparable amounts of proteinase K-resistant PrP$^{Sc}$ were measured in the brain of mice inoculated with either Fk-1C11 or APP$^{null}$-Fk-1C11 cells (Fig. 3c).

Finally, we showed that chronic intraperitoneal injection of the PDK1 inhibitor BX912 (5 mg per kg body weight per day; 0.25 μl h$^{-1}$) starting at 130 days post infection delayed mortality (Fig. 3a), reduced the motor deficits associated with prion infection (Fig. 3b) and decreased the PrP$^{Sc}$ level (Fig. 3c) with the same magnitude in mice infected with PrP$^{Sc}$ inocula Aβ-free or not. This indicates that the deregulation of PDK1 activity induced by prion infection is not influenced by Aβ species co-transmitted with PrP$^{Sc}$.

As a whole, these in vitro and in vivo data provide evidence that prion replication and infectivity are insensitive to the presence of Aβ in the inocula.

## PrP$^{Sc}$ increases CSF Aβ independently from Aβ co-transmission.

We further exploited PrP$^{Sc}$ inocula Aβ-free or not to address whether Aβ would fuel its own production when co-inoculated with PrP$^{Sc}$ in mice. To this end, we measured at

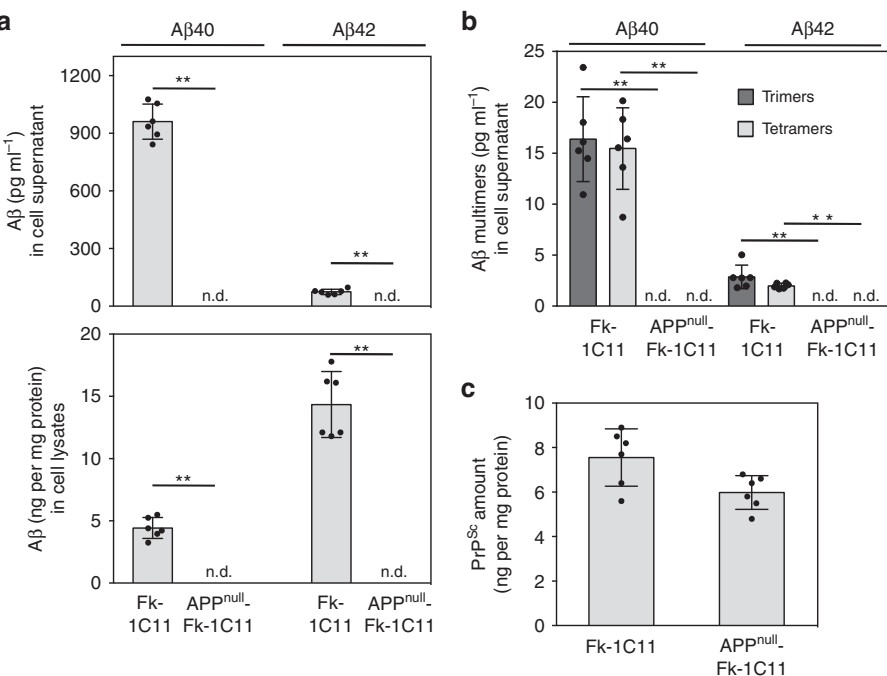

**Fig. 2** APP silencing does not affect PrP$^{Sc}$ replication in 1C11 cells. **a** Levels of Aβ40 and Aβ42 quantified by LC-MS/MS in the culture medium and lysates of Fukuoka-infected 1C11 cells expressing (Fk-1C11) or not (APP$^{null}$-Fk-1C11) the amyloid precursor protein APP. **b** ESI-IM-MS detection and quantification of Aβ40 and Aβ42 trimers and tetramers in the culture medium of Fk-1C11 and APP$^{null}$-Fk-1C11 cells. **c** ELISA-based quantification of proteinase K-resistant PrP$^{Sc}$ in Fk-1C11 and APP$^{null}$-Fk-1C11 cells. Values are means ± sem of six independent experiments. n.d. not detected. Data were analyzed using the two-tail Student t-test. $^{**}$denotes $p < 0.01$. Source data are provided as a Source Data file

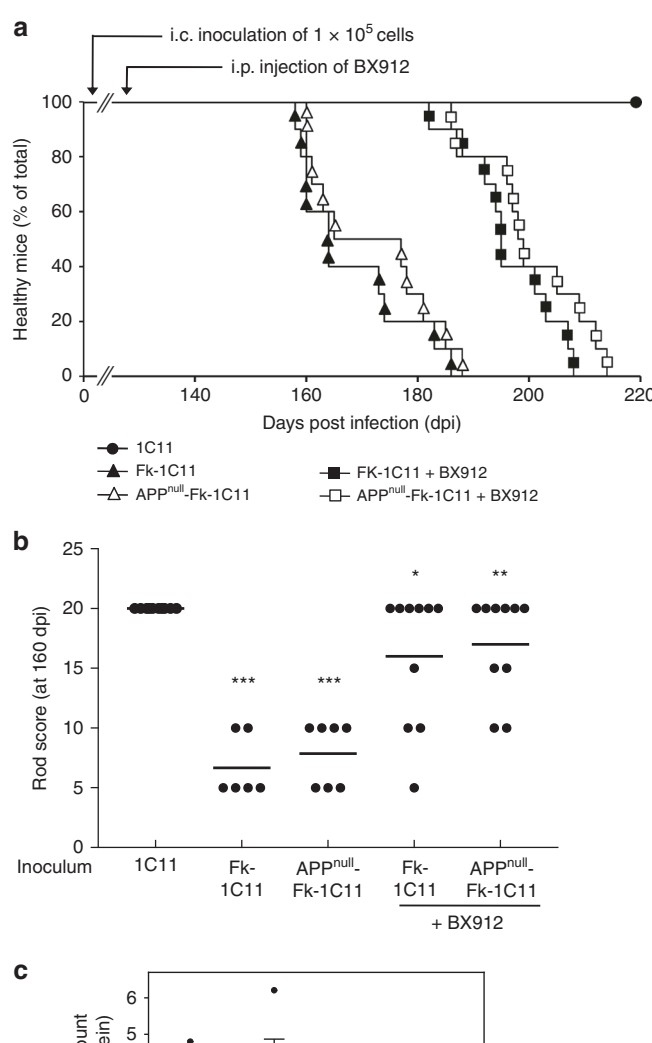

**Fig. 3** Aβ-free or not PrP$^{Sc}$ inocula similarly affect motor activity and survival of C57Bl/6 J mice. **a** Survival curves of C57Bl/6 J mice inoculated with $1 \times 10^5$ uninfected 1C11, Fk-1C11 or APP$^{null}$-Fk-1C11 cells via the intracerebral route (i.c.) infused or not with BX912 by intraperitoneal injection (i.p.) starting at 130 days after infection (5 mg per kg body weight per day; 0.25 μl h$^{-1}$; $n = 10$ per group). **b** Static rod test at 160 days after infection in mice inoculated with Fk-1C11 or APP$^{null}$-Fk-1C11 cells and treated or not with BX912 ($n = 6$–10 depending on the group) and in mice inoculated with uninfected 1C11 cells. **c** Post-mortem ELISA quantification of proteinase K-resistant PrP$^{Sc}$ in brains of C57Bl/6 J mice infected with Fk-1C11 or APP$^{null}$-Fk-1C11 cells-derived inocula and treated or not with BX912. Values are means ± sem. Data in Fig. 3b were analyzed using the two way ANOVA test with Bonferroni post-test correction. Data in Fig. 3c were analyzed using the two-tail Student $t$-test. ***denotes $p < 0.001$ vs. mice inoculated with uninfected 1C11 cells, *$p < 0.05$ vs. untreated Fk-1C11 injected mice, and **$p < 0.01$ vs. untreated APP$^{null}$-Fk-1C11 injected mice. Source data are provided as a Source Data file

160 dpi, when prion-infected mice start developing clinical signs, products of APP β-processing in the CSF of C57Bl/6 J mice inoculated intracerebrally with either Fk-1C11 or APP$^{null}$-Fk-1C11 cells.

As compared to mice injected with uninfected 1C11 cells, LC-MS/MS analyses showed highly similar increases in the levels of Aβ40, Aβ42 and sAPPβ in the CSF of mice inoculated with either Fk-1C11 cells (1.6-fold for Aβ40, 1.8-fold for Aβ42, and 2-fold for sAPPβ) or APP$^{null}$-Fk-1C11 cells (1.6-fold for Aβ40, 1.7-fold for Aβ42, and 2-fold for sAPPβ) (Fig. 4a–c). Since no change in prion infectivity was evidenced between PrP$^{Sc}$ inocula Aβ-free or not (Fig. 3a–c), we concluded that prion infection is sufficient to promote CSF accumulation of Aβ and sAPPβ, independently from the presence of any Aβ species in the inocula.

In the 160 dpi CSF of mice inoculated with Fk-1C11 cells, ESI-IM-MS analyses revealed the presence of Aβ40 and Aβ42 trimers (150.0 ± 3.7 and 33.0 ± 3.4 pg ml$^{-1}$, respectively) and tetramers (11.2 ± 2.2 and 11.5 ± 1.3 pg ml$^{-1}$, respectively) (Fig. 4e–f). Strikingly, in the 160 dpi CSF of mice inoculated with APP$^{null}$-Fk-1C11 cells, Aβ40 and Aβ42 tetramers (17.4 ± 5.3 and 23.8 ± 1.2 pg ml$^{-1}$, respectively) predominated, while Aβ40 and Aβ42 trimers (14.4 ± 2.2 and 2.8 ± 1.6 pg ml$^{-1}$, respectively) were ~10 times less abundant than in the CSF of mice inoculated with Fk-1C11 cells (Fig. 4e–f). Since Aβ trimers and tetramers were detected only in the CSF of mice infected with PrP$^{Sc}$ inocula Aβ-free or not, we concluded that prion infection is sufficient to promote the formation of Aβ multimers in vivo.

As for prion-infected 1C11$^{5-HT}$ cells, we recorded the PDK1-dependent imbalance of Aβ clearance ratios in the brain of mice inoculated with Fk-1C11 cells (Fig. 4g–h). Infection of mice with Fk-1C11 cells promoted a 1.8-fold and 1.5-fold increase of the CSF Aβ40 and Aβ42 production over degradation ratios, respectively, which were reversed upon chronic intraperitoneal injection of the PDK1 inhibitor BX912 (Fig. 4g–h). This indicates that in vivo PDK1 overactivation induced by prion infection directs APP towards the β-cleavage pathway. Accordingly, as compared to untreated infected mice, mice inoculated with either Fk-1C11 or APP$^{null}$-Fk-1C11 cells and infused with the PDK1 inhibitor exhibited at 160 dpi similar reductions in CSF Aβ40/42 and sAPPβ (Fig. 4a–c) and increases in CSF sAPPα (Fig. 4d) levels. In agreement with the BX912-mediated decreases of Aβ40/42, PDK1 inhibition in prion-infected mice also decreased CSF amounts of both Aβ40/42 trimers and tetramers (Fig. 4e–f).

These in vivo data thus show that the rise in mouse CSF Aβ40/42 and sAPPβ levels upon inoculation of APP expressing- or APP$^{null}$-Fk-1C11 cells depends on PrP$^{Sc}$ transmission only. The accumulation of Aβ40/42 monomers and the formation of Aβ40/42 multimers within a prion-infectious context originate from PrP$^{Sc}$-induced PDK1 overactivation. However, since the proportion of Aβ40/42 trimers and tetramers varies in the CSF of prion-infected mice depending on the presence or absence of Aβ species in PrP$^{Sc}$ inocula, this suggests that, in prion diseases, co-transmitted Aβ would influence the equilibrium between Aβ multimers generated upon prion infection.

**Aβ trimers boost the PrP$^{Sc}$-induced Aβ multimers production.** Among Aβ multimers, Aβ trimers self-propagate[32], display strong aggregative properties[33], and emerge as the elementary building blocks for the formation of supra-ordered Aβ multimers[34,35]. To assess whether Aβ trimers present in PrP$^{Sc}$ inocula would affect the production of Aβ40/42 multimers by prion-infected neurons, serotonergic 1C11$^{5-HT}$ neuronal cells were infected with PrP$^{Sc}$ homogenates prepared from $1 \times 10^5$ Fk-1C11 or APP$^{null}$-Fk-1C11 cells supplemented or not with a mixture of synthetic human Aβ40 (25 pg ml$^{-1}$) and Aβ42 (2.5 pg ml$^{-1}$) trimers. The

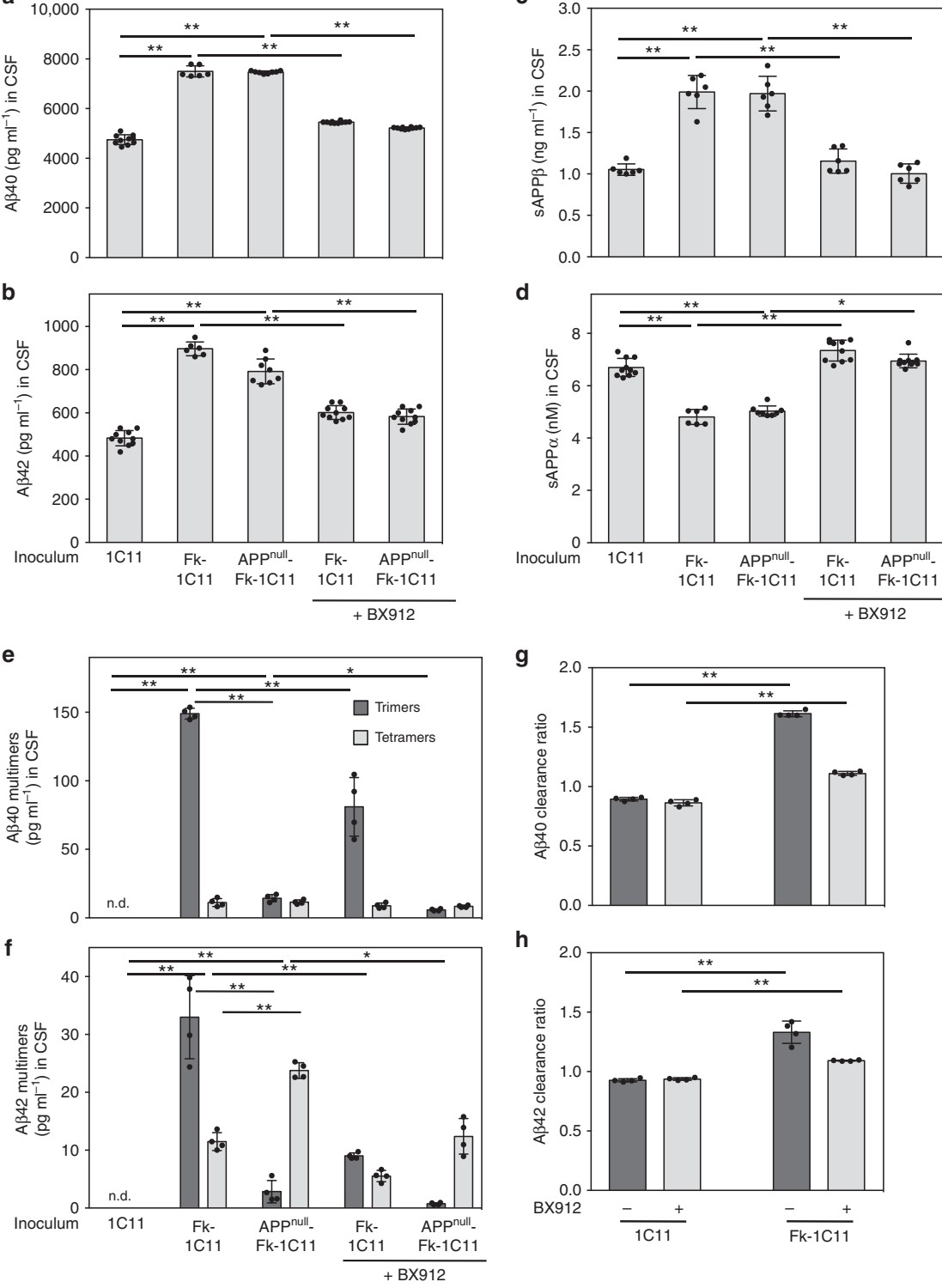

**Fig. 4** Deregulation of the PDK1-TACE axis in prion-infected C57Bl/6 J mice causes CSF accumulation of Aβ monomers and multimers. Concentrations at 160 dpi of Aβ40 (**a**), Aβ42 (**b**), sAPPβ (**c**), sAPPα (**d**), and multimers of Aβ40 (**e**) and Aβ42 (**f**) in the CSF of C57Bl/6 J mice inoculated with uninfected 1C11, Fk-1C11 or APP[null]-Fk-1C11 cells and infused or not with the PDK1 inhibitor BX912 ($n = 6$–10 depending on the group) deduced from LC-MS/MS (**a**–**d**) and ESI-IM-MS (**e**–**f**) analyses. **g**–**h** Aβ40 and Aβ42 clearance ratios measured in the CSF of mice inoculated with uninfected 1C11 or Fk-1C11 cells and infused or not with BX912 ($n = 5$). Values are means ± sem. n.d. not detected. Data in Fig. 4a–d were analyzed using the two way ANOVA test with Bonferroni post-test correction. Data in Fig. 4e–h were analyzed using the two-tail Student $t$ test. *denotes $p < 0.05$, and **$p < 0.01$. Source data are provided as a Source Data file

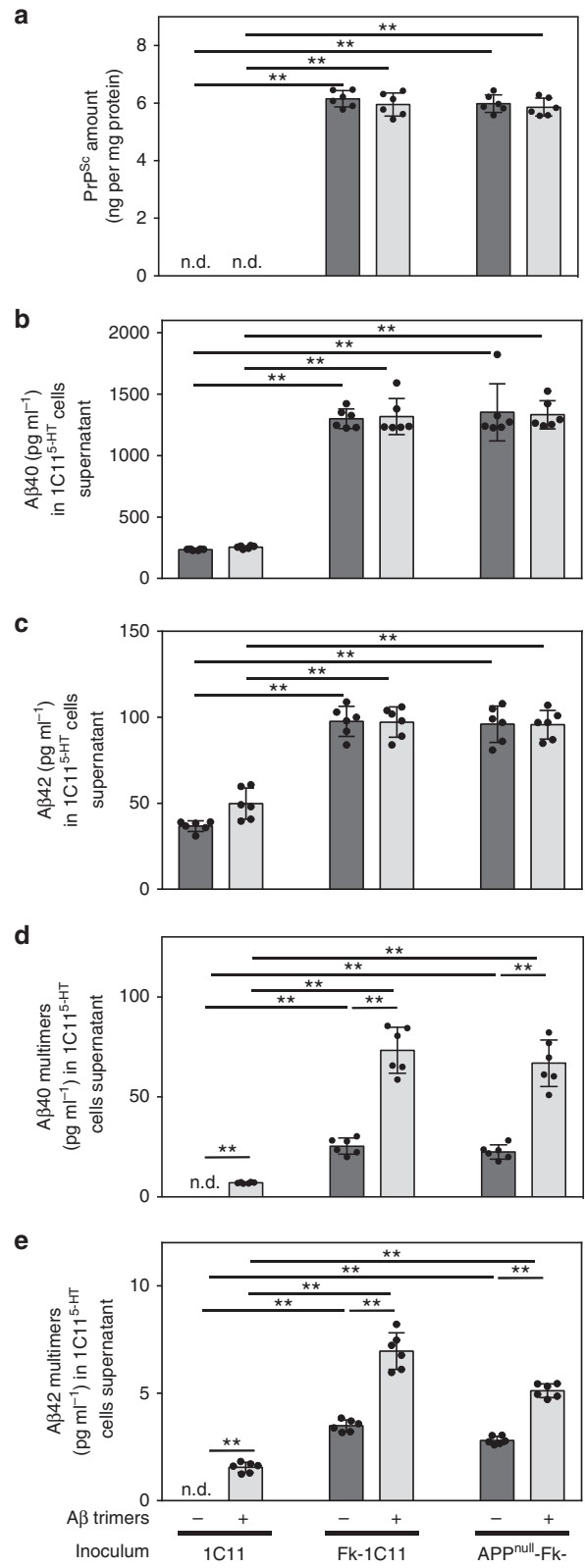

**Fig. 5** Addition of synthetic Aβ trimers to PrP$^{Sc}$ inocula amplifies the production of Aβ multimers. **a** Amount of PrP$^{Sc}$ in 1C11$^{5\text{-HT}}$ neuronal cells exposed to 1C11, Fk-1C11, or APP$^{null}$-Fk-1C11 cells-derived homogenates supplemented or not with synthetic Aβ trimers for 10 days, as assessed by ELISA after proteinase K digestion. Concentrations of Aβ40 (**b**), Aβ42 (**c**), and multimers of Aβ40 (**d**) and Aβ42 (**e**) in the cell culture medium of 1C11$^{5\text{-HT}}$ neuronal cells treated as in **a**, deduced from LC-MS/MS (**b, c**) and ESI-IM-MS (**d, e**) analyses. Values are means ± sem of six independent experiments. n.d. not detected. Data were analyzed using the two way ANOVA test with Bonferroni post-test correction. **denotes $p < 0.01$. Source data are provided as a Source Data file

Aβ trimers on basal neuronal Aβ production, independently from prion infection.

ELISA-based measurements of PrP$^{Sc}$ levels revealed that human Aβ40/42 trimers did not promote nor amplify PrP$^{C}$ conversion into PrP$^{Sc}$. Proteinase K-resistant PrP$^{Sc}$ was not detected in 1C11$^{5\text{-HT}}$ cells exposed for 10 days to uninfected 1C11 cell homogenates supplemented with Aβ trimers (Fig. 5a) and the amount of PrP$^{Sc}$ in 1C11$^{5\text{-HT}}$ cells infected with Fk-1C11 or APP$^{null}$-Fk-1C11 cell homogenates supplemented or not with Aβ trimers remained unchanged (Fig. 5a).

As concerns LC-MS/MS Aβ40/42 quantifications, addition of human Aβ trimers to uninfected 1C11 cell homogenates did not change the basal productions of Aβ40 (Fig. 5b) and Aβ42 (Fig. 5c) monomers by 1C11$^{5\text{-HT}}$ neuronal cells, indicating that Aβ trimers are unable to induce the production of Aβ40/42 monomers within a non-infectious context. Prion infection of 1C11$^{5\text{-HT}}$ cells with Fk-1C11 or APP$^{null}$-Fk-1C11 cell homogenates enriched or not with Aβ trimers triggered equivalent rises in Aβ40 (6-fold) and Aβ42 (2.5-fold) cell supernatant levels (Fig. 5b–c), showing that Aβ trimers present in the inocula do not interfere with the prion-stimulated production of Aβ40/42 monomers by neuronal cells.

As expected, Aβ40/42 trimers were detected by ESI-IM-MS in the supernatant of 1C11$^{5\text{-HT}}$ cells exposed to uninfected 1C11 cell homogenates supplemented with Aβ trimers (Fig. 5d–e). We however found lower concentrations of Aβ trimers ($7.1 \pm 0.3$ and $1.6 \pm 0.2$ pg ml$^{-1}$ for Aβ40 and Aβ42, respectively) than those introduced with the inoculum (25 pg ml$^{-1}$ for Aβ40 and 2.5 pg ml$^{-1}$ for Aβ42), suggesting a partial degradation of human Aβ40 and Aβ42 trimers after 10 days incubation with 1C11$^{5\text{-HT}}$ neuronal cells. Infection of 1C11$^{5\text{-HT}}$ cells with PrP$^{Sc}$ homogenates derived from either Fk-1C11 or APP$^{null}$-Fk-1C11 cells promoted comparable accumulations of Aβ40 ($25.4 \pm 3.7$ pg ml$^{-1}$ for Fk-1C11 vs. $22.5 \pm 3.3$ pg ml$^{-1}$ for APP$^{null}$-Fk-1C11) and Aβ42 ($3.5 \pm 0.2$ pg ml$^{-1}$ for Fk-1C11 vs. $2.8 \pm 0.2$ pg ml$^{-1}$ for APP$^{null}$-Fk-1C11) multimers in the surrounding milieu of prion-infected 1C11$^{5\text{-HT}}$ cells (Fig. 5d–e). This was ~50% amplified upon supplementation of PrP$^{Sc}$ inocula with human Aβ trimers ($73.4 \pm 10.5$ and $7.0 \pm 0.8$ pg ml$^{-1}$ for Aβ40 and Aβ42 multimers, respectively) (Fig. 5d–e) and exceeded the sum of the Aβ multimers concentrations produced upon prion infection plus Aβ trimers introduced in PrP$^{Sc}$ homogenates.

Again these in vitro data support the view that prion infection of neurons drives the production of Aβ40/42 monomers and multimers independently from the presence of Aβ in PrP$^{Sc}$ inocula. Although exogenous Aβ trimers have no impact on the production of Aβ40/42 monomers, Aβ trimers, likely acting as Aβ seeds, can boost the formation of Aβ multimers within a prion-infectious context.

**Aβ seeds transmitted with PrP$^{Sc}$ promote brain Aβ deposition.** The next question was to probe whether Aβ monomers and multimers generated upon prion infection would deposit and

concentrations of synthetic Aβ trimers added to PrP$^{Sc}$ homogenates were chosen by similarity to the Aβ trimer concentrations measured in the supernatants of chronically infected Fk-1C11$^{5\text{-HT}}$ cells (Fig. 1e–f). 1C11$^{5\text{-HT}}$ cells were also exposed to uninfected $1 \times 10^5$ 1C11 cell-derived homogenates supplemented or not with Aβ40/42 trimers in order to delineate the impact of

form Aβ plaques in the brain of prion-infected mice. As the presence of Aβ trimers in PrP$^{Sc}$ inocula influences the amount of Aβ multimers generated upon prion infection (Fig. 4e–f; Fig. 5d–e), we also assessed whether Aβ deposition would quantitatively vary according to the load of seeds of Aβ trimers co-transmitted with PrP$^{Sc}$ by exploiting PrP$^{Sc}$ inocula derived from Fk-1C11 and APP$^{null}$−1C11 cells supplemented or not with synthetic human Aβ trimers.

Because C57Bl/6 J mice do not deposit Aβ with age or when infused with Aβ-containing brain extracts[36], we used APP23 transgenic mice[37–39] as an experimental paradigm to probe the amyloidogenic properties of PrP$^{Sc}$ inocula. Three-month-old female APP23 transgenic mice were thus intracerebrally injected with PrP$^{Sc}$ inocula containing murine Aβ ($2.5 \times 10^5$ Fk-1C11 cells) or not ($2.5 \times 10^5$ APP$^{null}$-Fk-1C11 cells) enriched or not with synthetic human Aβ trimers (25 pg ml$^{-1}$ Aβ40 plus 2.5 pg ml$^{-1}$ Aβ42, $n = 10$ for each group). The capacity of synthetic Aβ trimers to promote Aβ deposition independently from prion infection was also challenged in APP23 mice injected with homogenates of $2.5 \times 10^5$ uninfected 1C11 cells supplemented with human Aβ trimers. Amyloid deposition in the mouse brain was followed by positron emission tomography (PET) imaging after [$^{11}$C]-Pittsburgh compound B (PiB) injection[23,40] from one to eight months post inoculation (mpi).

Mice injected with Fk-1C11 cells-derived inocula started depositing Aβ at 3 mpi and the number of mice with amyloid deposition progressively increased till 7 mpi (Fig. 6a, open triangles; Supplementary Table 1). Mice injected with APP$^{null}$-Fk-1C11 cells-derived inocula were not positive for Aβ deposits till 6 mpi (Fig. 6a, open squares; Supplementary Table 1). This suggests that PrP$^{Sc}$ does not display β-amyloidosis activity and that deposition of Aβ in APP23 mice injected with Fk-1C11 cells-derived inocula has been promoted by Aβ seeds co-transmitted with PrP$^{Sc}$. Accordingly, we showed that (i) supplementation of APP$^{null}$-Fk-1C11 cells-derived inocula with synthetic human Aβ trimers induced brain Aβ deposition as soon as 4 mpi (Fig. 6a, close squares; Supplementary Table 1), and (ii) enrichment of Fk-1C11 cells-derived inocula with synthetic human Aβ trimers accelerated Aβ deposition (start at 2 mpi) and enhanced the number of mice with amyloid deposition (Fig. 6a, close triangles; Supplementary Table 1). This supports the view that the Aβ trimers of murine or human origin present in PrP$^{Sc}$ inocula play the role of Aβ seeds responsible for amyloid deposition in prion-infected mice. None of the mice injected with homogenates of non-infectious 1C11 cells supplemented with synthetic human Aβ trimers deposited Aβ within a 8 month time-frame (Supplementary Table 1), indicating that, independently from prion infection, Aβ trimers fail to promote Aβ deposition.

We further provided evidence that the amount of brain Aβ deposits varied according to the load of Aβ trimers in PrP$^{Sc}$ inocula: robust depositions of Aβ occurred in APP23 mice injected with Fk-1C11 cells-derived inocula enriched with synthetic human Aβ trimers (Fig. 6a, close triangles), while lower Aβ depositions were recorded in mice inoculated with homogenates containing low levels of Aβ multimers, i.e., either Fk-1C11 cells-derived homogenates (Fig. 6a, open triangles) or APP$^{null}$-Fk-1C11-derived homogenates supplemented with synthetic human Aβ trimers (Fig. 6a, close squares). Corroborating the quantitative PET results, post-mortem histological analyses revealed that Aβ deposition in the brain of APP23 mice injected with Fk-1C11 cells-derived inocula supplemented with synthetic Aβ trimers was more substantial than in the brain of prion-infected APP23 mice that did not receive Aβ trimers (Supplementary Fig. 2A). We further showed that brain Aβ deposition in prion-infected APP23 mice injected or not with synthetic human Aβ trimers occurred mainly in the cortex, and to a lesser extent in

the thalamus, the hippocampus, the striatum, and the cerebellum (Supplementary Fig. 2B). This likely reflects the cortico-tropism of the Fukuoka-1 strain[41] and thereby the strongest incidence of prion infection on the production and deposition of Aβ in this brain area.

Importantly, we found that the survival of prion-infected APP23 mice depended on the amount of Aβ seeds co-transmitted with PrP$^{Sc}$. While APP23 mice infected with Fk-1C11-derived inocula (low level of Aβ seeds) died at >217 dpi (Fig. 6a, open triangles; Supplementary Table 1), APP23 mice infected with Fk-1C11-derived inocula supplemented with synthetic Aβ trimers (high level of Aβ seeds) displayed a reduced survival time ($180.3 \pm 0.4$ dpi) (Fig. 6a, close triangles; Supplementary Table 1). Enrichment of APP$^{null}$-Fk-1C11 inocula with Aβ trimers provoked the death of two mice out of ten (Fig. 6a, close squares; Supplementary Table 1), while none of the mice injected with prion inocula free of Aβ seeds died before they were sacrificed at 8 mpi (Fig. 6a, open squares; Supplementary Table 1). This suggests that brain Aβ deposition induced by Aβ trimers present in the inocula tends to accelerate the death of prion-infected mice. Since brain levels of PrP$^{Sc}$ were not statistically different between each prion-infected mice group at the end stage of the disease or when mice were sacrificed at 8 mpi (Fig. 6b), we concluded that the death of prion-infected APP23 mice does not relate to an over-accumulation of PrP$^{Sc}$ but is rather influenced by the induced Aβ pathology, whose severity depends on the dose of transmitted Aβ seeds.

These data thus indicate that prion infection generates conditions that permit brain Aβ deposition catalyzed by seeds of Aβ trimers co-transmitted with PrP$^{Sc}$. We showed that Aβ deposition induced by Aβ seeds in prion-infected APP23 mice appears to depend on PDK1, whose overactivation was strictly related to prion infection, but not to Aβ trimers (Fig. 6c). Infusion of the PDK1 inhibitor BX912 (5 mg per kg body weight per day; 0.25 µl h$^{-1}$, starting at 1 mpi) prevented Aβ deposition in APP23 mice injected with Fk-1C11 cells inocula (Fig. 6a, open circles; Supplementary Table 2) or delayed Aβ deposition in mice inoculated with Fk-1C11 cells- or APP$^{null}$-Fk-1C11 cells-derived homogenates supplemented with synthetic human Aβ trimers (Fig. 6a, close circles and close diamonds; Supplementary Table 2). BX912 infusion also significantly reduced PrP$^{Sc}$ levels (Fig. 6b), and thereby increased the survival time of all APP23 mice groups infected with prions (Fig. 6a, Supplementary Table 2). We thus concluded that the PDK1-dependent deregulation of APP α-processing upon prion infection fuels the production of Aβ40/42 to a certain threshold that allows cerebral β-amyloidosis induced by the transmitted exogenous Aβ seeds.

## Discussion

Our findings indicate that PrP$^{Sc}$-induced PDK1 overactivity promotes the production of Aβ40 and Aβ42 monomers and multimers both in vitro and in vivo. Whatever their structure, the Aβ peptides generated upon prion infection do not impact on PrP$^{Sc}$ replication nor take part to prion infectivity. We monitor that the Aβ peptides generated upon prion infection deposit in mouse brain only when PrP$^{Sc}$ and seeds of Aβ are co-inoculated to mice. Importantly, prion infection combined to brain Aβ plaque deposition accelerates the death of prion-infected mice.

Corruption of PrP$^C$ signaling function by PrP$^{Sc}$ is at the root of PDK1 overactivity via post-translational mechanisms. By stimulating the Src kinase-PI3K pathway, prion infection contributes to a rise in PDK1 activity[23]. This cascade promotes PDK1 translocation from the cytoplasm to the plasma membrane, docking PDK1 to PIP3 and inducing Src-mediated PDK1 phosphorylation at Tyr9 and Thr376 (ref. [42]). This gain of PDK1 activity within a

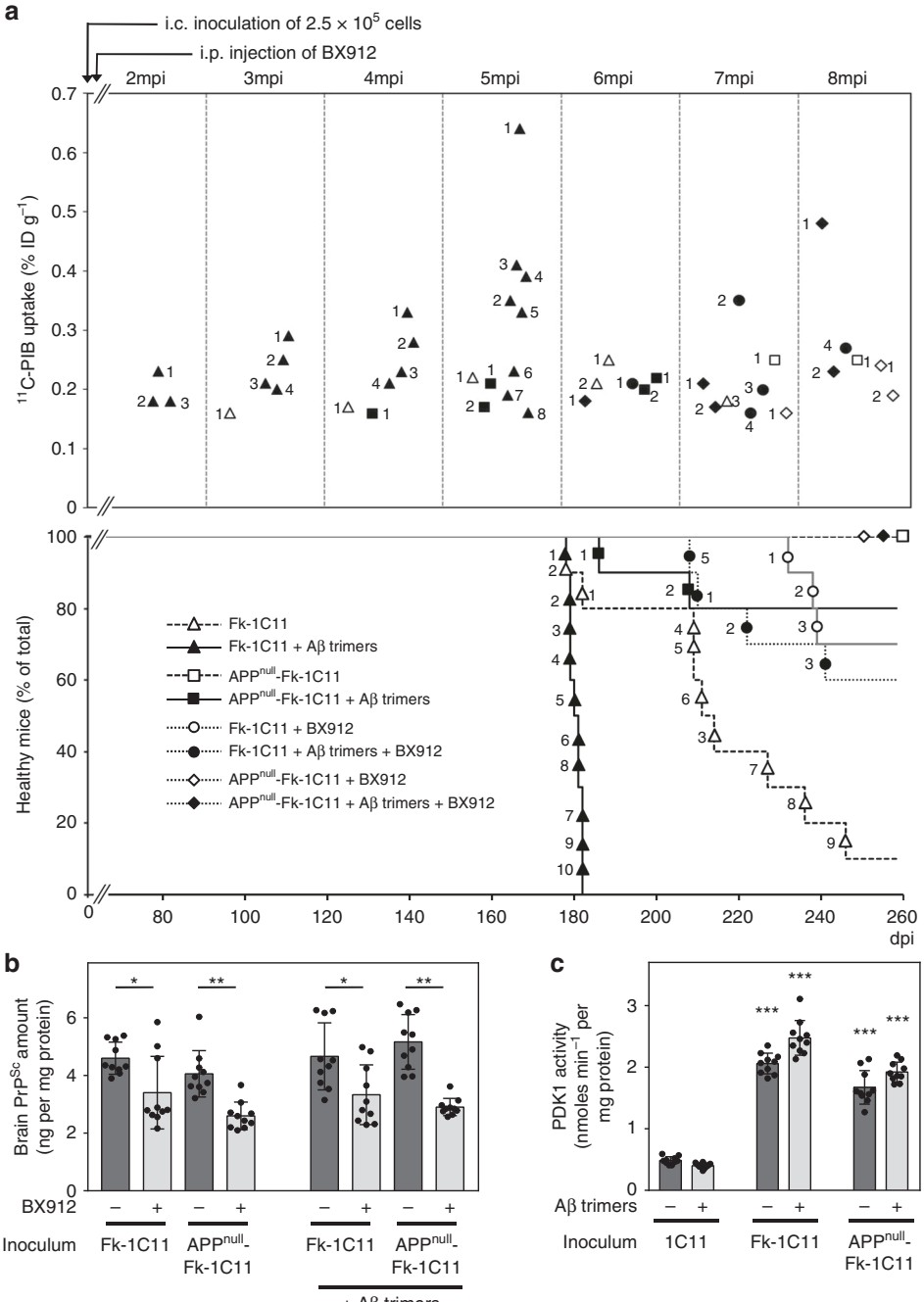

**Fig. 6** Aβ seeds co-transmitted with PrP$^{Sc}$ provoke brain Aβ deposition and decrease the survival of APP23 mice. **a** Time-dependent $^{11}$C-PIB uptake (% of the Injection Dose g$^{-1}$, upper panel) and survival curves (lower panel) of transgenic APP23 mice inoculated via the intracerebral route (i.c.) with 2.5 × 10$^5$ Fk-1C11 or APP$^{null}$-Fk-1C11 cells-derived homogenates supplemented or not with synthetic Aβ trimers, and infused or not with the PDK1 inhibitor BX912 (n = 10 per group). mpi months post inoculation, dpi days post inoculation. Each number refers to one mouse in each group. **b, c** ELISA-based quantification of Proteinase K-resistant PrP$^{Sc}$ amount (**b**) and PDK1 activity (**c**) in the brain of APP23 mice treated as in **a** at the end stage of prion disease or when sacrificed at 260 dpi. Values are means ± sem. Data in Fig. 6b–c were analyzed using the two way ANOVA test with Bonferroni post-test correction. *denotes $p < 0.05$, **$p < 0.01$, and ***$p < 0.001$ vs. APP23 mice inoculated with non-infectious 1C11 cells-derived homogenates supplemented or not with Aβ trimers. Source data are provided as a Source Data file

prion-infectious context also depends on the upstream over-activation of the ROCK kinase that interacts with PDK1 and phosphorylates PDK1 at Ser160 (ref. [24]). In prion-infected neurons, over-activated PDK1 then provokes the internalization of the α-secretase TACE in caveolin-1-enriched vesicles, reducing the TACE neuroprotective activity[23,24]. Internalization of TACE decreases its cleavage activity towards TNFα receptors (that accumulate at the plasma membrane and render diseased neurons

highly sensitive to TNFα inflammation) but also towards PrP$^C$ that amplifies the production of PrP$^{Sc}$ (ref. [23]). Our data provide evidence that PDK1-induced TACE internalization in prion-infected neurons or mice also attenuates the α-cleavage of APP into the neuroprotective sAPPα fragment and favors the APP β-processing into the neurotoxic Aβ40 and Aβ42 peptides. The pharmacological inhibition of PDK1 is sufficient to rescue sAPPα production to pre-infectious levels and to counteract Aβ40/42

accumulation. By showing a deficit of TACE-mediated APP α-processing within a prion-infectious context, our results extend the list of TACE substrates (whose products normally exert protective roles), which are no longer cleaved in prion-infected neurons. This may also concern the neurotrophin p75 receptor[43], for which a TACE-mediated shedding defect likely contributes to neuronal damages induced by the prion peptide 106–126 (ref. [44]).

Beyond an enhanced production of Aβ monomers, our data show that prion infection promotes the formation of Aβ40 and Aβ42 trimers and tetramers. In the cell culture medium of prion-infected neurons or in the CSF of prion-infected C57Bl/6 mice Aβ multimers represent ~2.75% of Aβ monomers and depend on the upstream production of Aβ monomers. The rescue of APP α-processing upon PDK1 inhibition and the subsequent down-regulation of APP β-processing reduce the level of Aβ multimers, reflecting a dynamic association equilibrium between Aβ species as observed in primary cultures of neurons from transgenic mice with Alzheimer-like pathology[45].

We found that Aβ peptides produced by prion-infected cells have no impact on prion replication and infectivity. This conclusion arose from our strategy, which consisted in silencing APP expression in 1C11 neuronal cells prior to prion infection. APP^null-1C11 cells no longer produce the Aβ40/42 peptides, but remain permissive to prion infection by the Fukuoka-1 strain. Importantly, the PrP^Sc level does not vary significantly between Fk-1C11 and APP^null-Fk-1C11 cells, clearly indicating that the Aβ produced upon prion infection do not contribute to PrP^C conversion into PrP^Sc. According to the concept that PrP^C transconformational changes depend on accessory proteins, i.e., protein(s) X[46–48], the invariance of PrP^Sc levels between prion-infected APP^null- and APP expressing 1C11 cells supports the view that neither APP nor its cleavage products are one of those proteins X facilitating PrP^Sc replication. With the help of Fk-infected APP^null-1C11 cells, we further established that the infectivities of prion inocula do not depend on the presence of Aβ. Intracerebral injections of 1C11 cells-derived PrP^Sc inocula Aβ-free or not in C57Bl/6 J mice indeed cause similar motor impairments, similar kinetics of a prion disease onset, and highly comparable accumulation of PrP^Sc in the brain. This latter point contrasts with the observation by Sarell et al. that synthetic soluble Aβ aggregates interfere with prion propagation in a cell-based prion bioassay, but only for Aβ concentrations in the micromolar range and higher[49]. Such Aβ concentrations are far to be reached within a prion-infectious context, according to the nanomolar range Aβ levels we found in the CSF of prion-infected C57Bl/6 J mice.

Further exploiting PrP^Sc inocula Aβ-free or not, we show that Aβ co-transmitted with PrP^Sc does not fuel the production of Aβ40 and Aβ42 monomers: the rise of monomeric Aβ40/42 in the culture medium of 1C11^5-HT neuronal cells and CSF of C57Bl/6 J mice is highly comparable whether the inocula contain Aβ or not. However, Aβ species present in PrP^Sc inocula are not neutral since (i) they influence the relative proportion of Aβ trimers and tetramers generated upon prion infection of C57Bl/6 J mice, and (ii) addition of Aβ40/42 trimers to PrP^Sc inocula amplifies the level of Aβ multimers produced by prion-infected neuronal cells.

Finally, we show that co-transmission of Aβ seeds with PrP^Sc is necessary to promote cerebral β-amyloidosis in prion-infected APP23 mice. Along prion pathogenesis, brain Aβ deposition does not occur in APP23 mice when injected with Aβ-free PrP^Sc inocula. Despite the capacity of prion infection to overstimulate the production of Aβ40/42 monomers and to generate Aβ multimers, PrP^Sc does not induce Aβ pathology, in agreement with Rasmussen et al.[50]. Aβ deposition within a prion-infectious context is caused by Aβ seeds co-transmitted with PrP^Sc. Of note, all APP23 mice inoculated with both PrP^Sc and synthetic human Aβ trimers deposit Aβ, while none of the mice only injected with the same dose of Aβ trimers display Aβ deposition, indicating that PrP^Sc and seeds of Aβ trimers cooperate for amyloid deposition. It is likely that PrP^Sc-stimulated production of Aβ monomers and multimers to a certain threshold generates favorable conditions for Aβ deposition induced by exogenous seeds of Aβ trimers. We also show that the amount of Aβ seeds co-injected with PrP^Sc determines the intensity of Aβ deposition: higher is the quantity of Aβ trimers in PrP^Sc inocula, higher and/or faster is Aβ deposition. This may reside in the capacity of seeds of Aβ trimers to amplify the PrP^Sc-dependent production of seedable Aβ multimers, to self-replicate and to aggregate. Our overall data would thus explain, at least in part, why Aβ deposition is not systematically associated with CJD cases of different etiologies in several age-matched patient cohort studies[51]. They would also account for the presence of Aβ plaques and Aβ pathology in the brain of patients who developed a iatrogenic Creutzfeldt-Jakob disease after injection of human growth hormone (hGH) contaminated with pathogenic prions[11] or after neurosurgical grafting of dura mater[13]. Demonstration has recently been made that PrP^Sc-contaminated hGH[52,53] as well as engrafted dura mater[13,54,55] were positive for Aβ seeds.

Most importantly, APP23 mice co-inoculated with PrP^Sc and synthetic Aβ trimers die before those injected with PrP^Sc only. As brain PrP^Sc levels are highly comparable between these two mice groups, an early death likely relates to the accumulation of Aβ multimers and deposition of Aβ in the brain of prion-infected animals. Severe cerebrovascular amyloid deposition, also called cerebral amyloid angiopathy (CAA), that is associated with marked Aβ deposition in capillaries, small arteries and arterioles, and vasculopathies, leads to hemorrhages and ischemic brain lesions (for review, see ref. [56], and references therein). The accelerated death of prion-infected mice with Aβ deposition might be due to the occurrence of mixed pathologies, that are prion disease and CAA[11]. In any case, inhibiting PDK1 counteracts the toxicity of both prion and transmitted Aβ seeds as BX912 infusion in prion-infected APP23 mice not only reduces the load of PrP^Sc, but also the brain deposition of Aβ, which thus reinforces the critical role of PDK1 in amyloid-based neurodegenerative diseases[23,24,57,58].

## Methods

**Chemicals.** Dibutyryl cyclic AMP (dbcAMP), cyclohexane carboxylic acid (CCA) were purchased from Sigma-Aldrich (St. Louis, MO, USA). The PDK1 inhibitor BX912 was from Axon MedChem BV (Groningen, The Netherlands). The TACE inhibitor, TNF-α processing inhibitor-2 (TAPI-2), was purchased from Peptides International (Louisville, KY, USA). The BACE1 inhibitor MK-8931 (Verubecestat) was from Abcam (Cambridge, UK).

**Preparation of synthetic human Aβ40/42 trimers.** Human Aβ40 and Aβ42 trimers were synthesized and purified in the Roche AG Chemistry Department by size-exclusion chromatography, reversed phase high-performance liquid chromatography, and cross-linking[28].

**Cell culture, APP silencing, and prion infection.** 1C11 precursor cells were established in the laboratory after stable transfection of mouse embryonal carcinoma F9 cells[59] with the recombinant pK4 plasmid[60,61]. 1C11 cells were grown in DMEM supplemented with 10% fetal calf serum and induced to differentiate along the serotonergic (1C11^5-HT) pathway[25]. Briefly, on addition of dbcAMP (1 mM) and CCA (0.05%), almost 100% of 1C11 cells acquire within 4 days a complete serotonergic phenotype (1C11^5-HT). To constitutively repress APP in 1C11 precursor cells, an shRNA coding sequence encompassing codons 498–504 of the *app* gene was introduced in the pTer plasmid[62] between the bglII and HindIII restriction sites under the control of the histone H1 promoter. The siRNA sequence targeting APP mRNA within the shRNA is 5'-UUGGCCAAGACAUCGUC GGdAdG-3' (refs. [31,63]). 1C11 cells were transfected by 2 μg pTer-shRNA/*app* plasmid[31] and clones silenced for APP expression were selected according to a reduction of APP level of more than 95% (referred to as APP^null-1C11 cells). 1C11,

APP[null]-1C11, or 1C11[5-HT] cells were chronically infected by the mouse-adapted Fukuoka (Fk) prion strain[26].

**Cell extract preparation**. Cells were washed in PBS/Ca$^{2+}$/Mg$^{2+}$ and incubated for 30 min at 4 °C in lysis buffer (50 mM Tris-HCl pH 7.4, 150 mM NaCl, 5 mM EDTA, 1% Triton X-100, protease and phosphatase inhibitors (Roche)). After centrifugation of the lysate (14,000 × g, 15 min), the protein concentration in the supernatant was measured with the bicinchoninic acid method (Pierce, Rockford, IL, USA).

**sAPPα, sAPPβ, Aβ40, and Aβ42 quantification**. The detection and quantification of the different APP cleavage products (sAPPα, sAPPβ, Aβ40, and Aβ42) in the culture medium and lysates of prion-infected cells as well as in the CSF of prion-infected mice were achieved by a stable-isotope dilution methodology in combination with LC-MS/MS[64].

Detection and quantification of Aβ multimers were performed by ESI-IM-MS[28]. Approximately 50 µl of cell culture supernatant or CSF were added to 950 µl of methanol and acetic acid (1%) solution maintained at 37 °C for 30 min and then centrifuged for 1 h at 1500 × g. The supernatant was subjected to analysis by ESI-IM-MS without further sample preparation. Electrospray Ionization Mass Spectrometry (ESI-MS) coupled with Traveling Wave Ion Mobility Spectrometry (TWIMS) experiments were performed with a Synapt G2 HDMS mass spectrometer (Waters) equipped with a Z-spray ESI source. Capillary voltage of 2.00 kV, sampling cone voltage of 50 V, extraction cone voltage of 4.0 V, source temperature of 100 °C, desolvation temperature of 200 °C, trap and transfer collision energy of 10 eV were used as parameters for ESI. Although no signal was detected over 3000 m/z, data were acquired over the m/z range of 500–5000 for 2 min. Helium flow rate of 30 ml min$^{-1}$ and nitrogen flow rate of 25 ml min$^{-1}$ were used for TWIMS experiments, with 250 m s$^{-1}$ for wave velocity and 8.0 V for wave height. Calibration of the experimental arrival time to determine collision cross section ($\Omega D$) values was performed following previously reported methods[65,66] using ubiquitin and cytochrome c as calibrants. The raw data were processed using Mass Lynx v4.1 software (Waters). The reported data are the average of three independent experiments. The ESI-IM-MS data were assigned to specific Aβ oligomers on the following basis: (i) First, ion mobility is often described as proportional to collision cross section-to-charge ratio ($\Omega/z$). Any charge state (m/z) in the mass spectrum could correspond to several species (e.g., M + z, D + 2z, Tr + 3z, Te + 4z; with M = monomers, D = dimers, Tr = trimers, and Te = tetramers). These species have the same m/z value and are therefore indistinguishable in the mass spectrum. They can be distinguished by IM because their $\Omega/z$ differ. For example, the Te charge would be theoretically four times that of M; however, the $\Omega Te$ would be smaller than four times the $\Omega M$ because of the favorable interactions between the monomers and higher order species. Thus $\Omega Te/4z < \Omega Tr/3z < \Omega D/2z < \Omega M/z$, i.e., the species that have smallest drift times were initially assigned to those corresponding to the largest oligomer. For the Aβ40 spectra at m/z 2165, four contributions were detected in the drift time domain at 4.1, 5.1, 7.7, and 12.8 ms. The peaks at 4.1 and 12.8 ms were initially assigned to the highest and lowest oligomer order, respectively. (ii) Second, we assigned a given oligomer order on the basis of the $^{13}$C isotope distribution associated with each of the peaks separated in the mobility dimension[67]. Applying this consideration to the m/z peak of Aβ40 observed at m/z 2165, the $^{13}$C isotope distribution associated with the mobility peaks detected at 12.8 and 7.7 ms were consistent with charges + 2 and + 4, respectively, and therefore were assigned to M + 2 and D + 4, respectively. The resolution of the $^{13}$C isotope distribution for the mobility peaks observed at 5.1 and 4.1 ms was not sufficient for their assignation. In this case, the charge envelope in the 2D ESI-IM-MS spectrum was analyzed. In the ESI-IM-MS spectra, we found other charge states consistent with the assignment of the peaks at 5.1 and 4.1 ms to Tr + 6 and Te + 8, respectively. For the former, these were Tr + 5 and Tr + 7 and for the latter Te + 7. (iii) Third, the conformation of the ions was also taken into account. Peaks that had the same m/z and the same $^{13}$C isotope distribution were attributed to compact and extended forms of the same oligomer. The m/z peak of Aβ40 observed at m/z 1732 showed two contributions in the drift time domain (at 4.4 and 7.4 ms), and the $^{13}$C isotope distribution associated with these two mobility peaks was consistent with +5 charges, so they were assigned to compact and extended forms of the D + 5, respectively. Finally, the limits of detection (LOD) of Aβ40/42 monomers and oligomers by ESI-IM-MS were as followed: 0.09 pg ml$^{-1}$ for Aβ40/42 monomers, 0.10 pg ml$^{-1}$ for dimers, 0.12 pg ml$^{-1}$ for trimers, 0.16 pg ml$^{-1}$ for tetramers, 0.20 pg ml$^{-1}$ for pentamers, 0.25 pg ml$^{-1}$ for hexamers, 0.28 pg ml$^{-1}$ for heptamers and 0.29 pg ml$^{-1}$ for octamers.

**PrP$^{Sc}$ quantification**. The amounts of proteinase K-resistant PrP$^{Sc}$ in Fk-infected 1C11, 1C11[5-HT], and APP[null]-1C11 cell extracts as well as in brain extracts were determined using a PrP-specific sandwich ELISA[68] after proteinase K digestion (10 µg ml$^{-1}$) for 1 h at 37 °C.

**Ethics statement**. Adult C57Bl/6 J mice and transgenic APP23 mice were bred and underwent experiments in level-3 biological risk containment, respecting European guidelines for the care and ethical use of laboratory animals (Directive 2010/63/EU

of the European Parliament and of the Council of 22 September 2010 on the protection of animals used for scientific purposes). Ten mice per group were inoculated intracerebrally with 20 µl of sample containing cell extracts ($1 \times 10^5$ cells for 8-week-old male C57Bl/6 J mice; $2.5 \times 10^5$ cells for 3-month-old female APP23 mice)[69]. Cells were submitted to three freeze-thaw cycles and suspensions were sonicated for 2 min (Cup-horn sonicator; Nanolab Inc, Waltham, MA, USA). CSF was collected from the cisterna magna under anesthesia with 3% isoflurane. All animal procedures were approved by the Animal Care and Use Committee at Basel University (Switzerland) and by the Comité Régional d'Ethique en Matière d'Expérimentation Animale de Strasbourg (France) with number CEEA35 ref AL/01/01/01/13.

**Chronic intraperitoneal injection of BX912 into mice**. Mice were fasted overnight but allowed water ad libitum before the experiment. They were then anesthetized with isoflurane inhalation, and a midline incision was performed to insert into the peritoneum the polyethylene catheter of an osmotic pump (Alzet, Cupertino, CA, USA). BX912 (PDK1 inhibitor) or vehicle (1% DMSO in sterile normal saline buffer) was administered at a flow rate of 0.25 µL per h, which corresponded to 100 µg per mouse per day (5 mg kg$^{-1}$ per day). Pumps were replaced every 4 weeks.

**Behavioral testing**. Motor function in prion-infected mice was assessed by the static rod test[70].

**Measurement of Aβ production over degradation ratios**. 1C11[5-HT] and their infected counterparts were incubated with $^{13}$C$_6$-leucine (98% $^{13}$C$_6$) in OptiMEM medium for 6 h. Pulse medium was then replaced by fresh OptiMEM and supernatants were collected over a 12 h time window. Aβ42 and then Aβ40 were serially immunoprecipitated from the samples using C-terminal specific antibodies (NB300–225 from Novus Biologicals and ABIN363343 from Antibodies online, Atlanta, GA, USA). Purified Aβ peptides were then digested with trypsin and $^{13}$C$_6$-leucine abundance in these tryptic fragments quantified using tandem mass spectrometry[29,30]. For in vivo measurement of Aβ clearance ratios, the same methodology was used except that $^{13}$C$_6$-leucine was delivered intravenously by a miniosmotic pump model 2002 (Alzet, Cupertino, CA).

**In vivo detection of amyloid deposition**. Amyloid deposition in transgenic APP23 mice were detected by PET imaging[40] using a microPET FOCUS F120 scanner (Siemens Medical Solutions, Bern, Switzerland). The Pittsburgh B compound was from Scintomics (Fuerstenfeldbruck, Germany) and used at a concentration of 1.5 nmol l$^{-1}$. The threshold of PET positivity was considered as the limit of quantitation (LOQ) of PIB binding determined by the statistical method developed in ref. [71]. Briefly, according to guidelines of the Clinical and Laboratory Standards Institute (CLSI EP17-A2)[72], PIB bindings were performed on four control mice (two females and two males, three months-old, free of Aβ deposits as assessed by histology), three replicates a week for 5 weeks with two different lots of PIB, i.e., 120 replicates in total. The limit of detection (LOD) calculated by the adjusted Currie method[73] was 0.05 ± 0.01 % (SD). According to the statistical method of LOQ determination, LOQ is 10 SD above LOD, resulting that PIB uptake was considered significant for values superior to 0.15% of the injected dose (ID) per g of animal.

After µPET scans, animals were sacrificed, and brain tissue was fixed, paraffin-embedded and cut into 5-µm coronal sections. De-waxed and rehydrated sections underwent tissue depigmentation in potassium permanganate, oxalic acid and water and were subsequently pretreated with 70% formic acid for 10 min for epitope retrieval. Endogenous peroxidase activity was blocked by rinsing in 3% hydrogen peroxide for 5 min. Sections were thereafter incubated for 1 h with anti-amyloid primary 4G8 antibody (SIG-39220, dilution 1:500) from Covance (Princeton, NJ, USA), HRP-labelled secondary antibody (anti-mouse, 30 min) and visualized with diaminobenzidine (DAB). Finally, the sections were counterstained with haematoxylin, dehydrated by submerging in a series of alcohol, fixed in xylene, mounted and coverslipped. Virtual images were acquired using a Mirax Digital Slide Scanner (Zeiss), and image analysis was performed using the Definiens analysis software package v1.5. Regions of interest (ROIs) were manually delineated in accordance with Franklin and Paxinos atlas, and for each ROI, the percentage of DAB-labelled area was calculated.

**Measurement of PDK1 activity**. PDK1 activity was measured in cell extracts using a fluorescently labeled PDK1 substrate (5FAM-ARKRERTYSFGHHA-COOH, Caliper Life Sciences, Hanover, MD, USA)[74]. The relative amounts of substrate peptide and product phospho-peptide were determined using a Caliper EZ-reader (Caliper Life Sciences, Hanover, MD, USA).

**Data analysis**. An analysis of variance of the cell/animal response group was performed using the Kaleidagraph software (Synergy Software, Reading, PA, USA) and the GraphPad Prism software (San Diego, CA, USA). Values are given as means ± s.e.m. Significant responses ($P < 0.05$) and their corresponding P-values are provided in figure legends. Survival times were analyzed by Kaplan-Meier survival analysis using a log-rank test for curve comparisons.

**Reporting summary**. Further information on research design is available in the Nature Research Reporting Summary linked to this article.

## Data availability

The data that support the findings of this study are available from the corresponding authors upon reasonable request. The source data underlying Figs. 1–6 are provided as a Source Data file.

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

## Acknowledgements

We thank V. Mutel, G. Zürcher, E. Borroni, Z. Lam, M. Bühler, N. Pierron, and R. Hochköppler for providing skillful methodological assistance for all data acquisition and statistical analyses. We acknowledge M. Briley for helpful discussions and critical reading of the manuscript. This work was supported by the French Agence Nationale de la Recherche (TargetingPDK1inAD, n°ANR-16-CE16–0021–01), the European Joint Program on Neurodegenerative Diseases and Agence Nationale de la Recherche (PrP$^C$&PDK1, n° ANR-14-JPCD-0003–01), the foundation Vaincre ALZHEIMER (grant n°FR-15033) and INSERM. JE is funded by Domaine D'Intérêt Majeur - Cerveau&Pensée - Région Ile de France. ZAA is a post-doctoral fellow funded by the ANR PrP$^C$&PDK1 European project and the ANR TargetingPDK1inAD program. VB is funded by the ANR PrP$^C$&PDK1 European project.

## Author contributions

Conceived and designed the experiments: J.M.L., O.K. and B.S. Performed the experiments: J.E., V.B., Z.A.A., F.B.D., A.M.H., Y.B. and J.M.L. Analyzed the data: J.E., V.B., M.P., A.B., O.K., J.M.L. and B.S. Wrote the paper: M.P., A.B., O.K., J.M.L. and B.S.

## Additional information

**Competing interests:** J.M.L has non-financial competing interests with Hoffmann La Roche Ltd laboratories. He acts as an expert witness for Hoffmann La Roche Ltd laboratories. This does not alter our adherence to all Nature Communications policies on sharing data and materials. The remaining authors declare no competing interests.

