## [Peer Review File with Redactions · Nature Communications]

Editorial Note: The name of Reviewer #1 has been removed by request on p1 and p4 of this peer review file.

Reviewers' comments:

Reviewer #1 (Remarks to the Author):

The authors have studied the interplay between PrPSc, Abeta species, PDK1, and TACE in vitro and in vivo. They report evidence that PrPSc-enhanced PDK1 activity and resultant decrease in TACE activity leads to higher concentrations of Abeta monomers and oligomers, but that the latter do not impact prion replication or infectivity. It is interesting and significant that the authors find no evidence that inoculation of PrPSc alone leads to Abeta amyloid accumulation. However, if trimeric Abeta seeds are added to a PrPSc inoculum, then Abeta amyloid accumulation can occur. Recent evidence of Abeta pathology in iatrogenic CJD patients receiving Abeta aggregate-containing growth hormone injections makes this work particularly timely. However, I have some suggestions to improve the manuscript.

1. It would be helpful if the authors more carefully indicated when their experimental systems involve murine vs human Abeta sequences.
2. The manuscript would benefit from a more careful examination of English usage.
3. More extensive explanations of the mechanism by which PrPSc enhances PDK1 activity would be helpful, whether determined in previous studies, or added to the current one. For example, is PDK1 mRNA enhanced?
4. L132-133: Is the MS technique capable of detecting higher order oligomers with similar sensitivity?
5. In the in vivo experiments in which PrPSc levels are compared, were all of these measurements made at end stage, regardless of the relative survival times, or at comparable dpi's? Should be specified.
6. PET analyses, especially as portrayed in Fig 6A: These analyses and data presentation were difficult to understand for the uninitiated. Please provide more explanation of what the data points represent and how the threshold of positivity was determined. An indication of signals from negative controls, and their variability needs to be provided.
7. Also, following on the previous concern, additional types of measurements of Abeta deposition (such as neurohistological analyses) should be provided to augment the PET results.
8. L333-336: This conclusion is not convincing. It would seem that too few mice getting sick in either group to make much of a statement like this.
9. L336-339: Following on point #5 above, were these measurements all made at the clinical end point? It seems to me that to better support this conclusion, one would need to compare animals at same time point, e.g. at ~180 dpi for +/- trimers with Fk-1C11 inoculum.

[REDACTED]

Reviewer #2 (Remarks to the Author):

Amyloid plaques containing the Alzheimer's disease-related A β peptide can be found in human prion diseases such as Creutzfeldt-Jakob disease (CJD). However, it's relevance to prion disease progression and pathogenesis is unknown. The manuscript by Ezpeleta et al. uses a prion-infected cell line that does or does not express the Alzheimer's precursor protein (APP) to study the possible connection between prion pathogenesis and A β deposition. They show that APP processing by specific proteases is altered during prion infection, in vitro and in vivo, leading to increased production of A β monomers and trimers. Using a transgenic mouse model for A β deposition, they further show that the presence of A β trimers in the prion inoculum leads to A β deposition and more rapid prion disease. They conclude that alterations in A β processing by proteases leads to increased levels of A β deposition which can accelerate prion disease.

While the paper can be somewhat difficult to follow, the data are clearly presented and support the conclusion that PDK1 dysregulation during prion infection can lead to increased levels of A β in

tissue culture cells as well as in a transgenic mouse line used to model certain aspects of Alzheimer's disease. However, the manuscript suffers somewhat from overinterpretation of the data. For example, they state that PrPSc induces overactivation of PDK1 and is sufficient to promote elevated levels of A β in cerebral spinal fluid (lines 228-230 and 200-202). However, only complex prion homogenates are being tested. There are no data showing that PrPSc is directly mediating either of these effects. To draw this conclusion, experiments utilizing purified PrPSc would have to be done. The authors need to be more accurate in how they state their conclusions so that they match the data presented.

- 1) In Figure 1E and F, what percentage of the total A β in the cell supernatant of prion infected cells is oligomeric, i.e. trimers or tetramers? In the cell lysate, only 0.2% of the A β was oligomeric (line 136). Is a similarly low percentage of A β oligomeric in the supernatant?
- 2) Why weren't uninfected 1C11 cells treated with BX912 and TAPI-2? Wouldn't this be a control for how PDK1 and TACE react to their respective inhibitors in the absence of prion infection?
- 3) No methods are given for how the oligomeric state of A β was determined using ion mobility and EIS-IM-MS (lines 1250126). How did they control for the possibility that oligomerization of A β occurred during and/or after preparation of the sample as opposed to within the prion-infected cell or brain?
- 4) How do the authors explain the somewhat paradoxical result that, when compared with prion-infected cells that did express APP, much higher levels of A β tetramers were found in mice inoculated with prion-infected cells that did not express APP? It's not entirely clear from the results how they are interpreting this result.
- 5) If available, it would be informative to have either the PET scans for the data in Figure 6A or histopathological confirmation of A β deposition in the APP23 mice infected with prions. While the data in Figure 6A give a quantitative measurement of A β , it would be of interest to know the regions of the brain where A β has deposited as amyloid.
- 6) In line 250, the term "PrPres" is used. Does this refer to the PrPSc data in Figure 5A? If so, the terms should match.

Reviewer #3 (Remarks to the Author):

The authors investigate functional aspects of the observed Abeta histopathology in infectious prion disease. This is especially interesting given the emerging relationship between PrP and Abeta in Alzheimer's disease (AD). Specifically, the authors examine how Abeta peptides derived from prion-infected cells affect prion infectivity (they don't), how prion infection affects Abeta production (PDK1-TACE dependently), and how Abeta influences prion-dependent mortality (accelerates it).

Overall the study feels incomplete in each of the areas of investigation. And while the role of Abeta in prion disease is an interesting and important topic, the insights here are incremental and somewhat fragmentary in their demonstration.

The oligomeric state of the Abeta in this study is poorly characterized. This is important because the predominant PrP-interacting species of Abeta oligomer in both human AD brain and mouse AD model brain are high molecular weight aggregates in the 100-mer range, observable via size exclusion chromatography or native PAGE. The authors declare no higher order oligomers of Abeta (above tetramer) were evidenced in samples, but show no data to support that claim. On what basis is this conclusion reached? The presence of Abeta trimers and tetramers in the absence of

larger oligomers would be notable and requires thorough demonstration through more than ESI-IM-MS. In general it would be important to show the overall similarity of all oligomeric species for brain extracts, CSF, and cell-based inocula.

The study does not address how Abeta exerts its toxic action. Is it via PrPc-binding as in AD? This should be explored.

In figure 1, it is difficult to interpret the relationship between TACE and PDK1 because there is no condition where the TACE TAPI-2 is applied alone to FK-1C11 cells. It would also be informative to show here whether TAPI-2 and BX912 have an effect on 1C11 cells in the absence of prion infection. Is the role of TACE and PDK1 completely prion dependent?

The unconventional use of asterisks denoting statistical significance is confusing. The use of one asterisk (*) to denote $p < 0.001$ with two asterisks (**) to denote $p < 0.01$, is inverted from convention.

The conclusion that the beta secretase pathway is being favored by PrPsc seems likely but needs to be directly demonstrated with manipulation of beta secretase genetically or pharmacologically.

The assertion that Abeta trimers, likely acting as seeds, can boost the formation of Abeta multimers within a prion infectious context implies that the prion infection enables the action of Abeta trimers. This would be better supported if the action of Abeta trimers were examined outside the context of prion infection. Are they able to boost multimers in the absence of prion infection or is there interaction between the trimers and prion? AD model mice can have accelerated mortality. Are trimers able to accelerate death independently of prion infection, and thus suggest additive parallel mortality-enhancing pathways? Or does the mortality enhancement of trimers depend on prion co-infection, suggesting mechanistic convergence? Without examining the effect of Abeta independently, it is difficult to draw hard conclusions.

It would be advisable for the authors to enlist the services of an editor with English as a first language. In places, grammatical peculiarities make interpretation challenging.

Reviewer #1 (Remarks to the Author):

The authors have studied the interplay between PrP^{Sc}, Abeta species, PDK1, and TACE in vitro and in vivo. They report evidence that PrP^{Sc}-enhanced PDK1 activity and resultant decrease in TACE activity leads to higher concentrations of Abeta monomers and oligomers, but that the latter do not impact prion replication or infectivity. It is interesting and significant that the authors find no evidence that inoculation of PrP^{Sc} alone leads to Abeta amyloid accumulation. However, if trimeric Abeta seeds are added to a PrP^{Sc} inoculum, then Abeta amyloid accumulation can occur. Recent evidence of Abeta pathology in iatrogenic CJD patients receiving Abeta aggregate-containing growth hormone injections makes this work particularly timely.

We thank [REVIEWER #1 FOR THEIR] nice appreciation of our work.

However, I have some suggestions to improve the manuscript.

1. It would be helpful if the authors more carefully indicated when their experimental systems involve murine vs human Abeta sequences.

Aβ40 and Aβ42 peptides produced by 1C11 cells, their serotonergic 1C11^{5-HT} neuronal derivatives and C57Bl/6J mice are murine Aβ.

Since APP23 mice overexpress mutated human APP, we used synthetic Aβ40/42 trimers of human origin (as in ref 27) to supplement cell-derived homogenates infected or not by prions.

To improve the clarity of our manuscript, the human origin of synthetic Aβ trimers is now indicated in the manuscript.

2. The manuscript would benefit from a more careful examination of English usage.

Native English-speaking people read the manuscript. This new version of our manuscript takes into account all their grammar corrections.

3. More extensive explanations of the mechanism by which PrP^{Sc} enhances PDK1 activity would be helpful, whether determined in previous studies, or added to the current one. For example, is PDK1 mRNA enhanced?

Mechanisms sustaining PDK1 over-activation in prion-infected neurons were described in our previous studies published in Nature Medicine (see Pietri et al., 2013) and PloS Pathogens (see Alleaume-Butaux et al., 2015). In those studies, we showed that PrP^{Sc} stimulates PDK1 activity through post-translational mechanisms: (i) PrP^{Sc} activates the Src kinase-PI3 kinase pathway (Nat. Med., 2013) that produces the PIP3 phospholipid. PIP3 serves as a docking platform for PDK1 at the plasma membrane. Translocation of PDK1 from the cytosol to the plasma membrane was shown to increase PDK1 activity. Then Src kinases bind PDK1 and phosphorylate PDK1 at Tyr9 and Thr376, contributing to a rise in PDK1 activity [1]. (ii) Our study in Plos Pathogens (2015) shows that PrP^{Sc} overactivates another kinase, ROCK, that binds to PDK1 and phosphorylates PDK1 at Ser160. This additional phosphorylation step contributes to a gain of PDK1 activity within a prion-infectious context.

No variation of PDK1 protein level was evidenced in prion-infected neurons as compared to non-infected neurons (see Alleaume-Butaux *et al.*, PloS pathogens, 2015).

The post-translational mechanisms by which PrP^{Sc} provokes a rise in PDK1 activity are now introduced in the discussion section: "Corruption of PrP^C signaling function by PrP^{Sc} is at the root of PDK1 overactivity via post-translational mechanisms. By stimulating the Src kinase - PI3K pathway,

prion infection contributes to a rise in PDK1 activity (Pietri et al., Nat. Med, 2013). This cascade promotes PDK1 translocation from the cytoplasm to the plasma membrane, docking PDK1 to PIP3 and inducing Src-mediated PDK1 phosphorylation at Tyr9 and Thr376 (Calleja, 2014 [1]). This gain of PDK1 activity within a prion-infectious context also depends on the upstream over-activation of the ROCK kinase that interacts with PDK1 and phosphorylates PDK1 at Ser160 (Alleaume-Butaux et al., Plos Pathogens, 2015).” (page 18).

[1] Calleja V, Laguerre M, de Las Heras-Martinez G, Parker PJ, Requejo-Isidro J, Larijani B. Acute regulation of PDK1 by a complex interplay of molecular switches. *Biochem Soc Trans.* 2014 Oct;42(5):1435-40. Review.

4. L132-133: Is the MS technique capable of detecting higher order oligomers with similar sensitivity?

According to the *in vitro* study by Pujol-Pina et al. (Scientific Reports, 2015), the ESI-IM-MS technique allows to discriminate monomers to hexamers of A β 40 and A β 42 in a preparation of pure synthetic human A β 40 and A β 42. In that study, pentamers and hexamers could be detected and quantified even if they represented ~3% and ~1% of total A β species, respectively.

In our hands, the limits of detection (LOD) of A β monomers and oligomers by the ESI-IM-MS technique was 0.09 pg ml⁻¹ for A β 40/42 monomers, 0.10 pg ml⁻¹ for dimers, 0.12 pg ml⁻¹ for trimers, 0.16 pg ml⁻¹ for tetramers, 0.20 pg ml⁻¹ for pentamers, 0.25 pg ml⁻¹ for hexamers, 0.28 pg ml⁻¹ for heptamers and 0.29 pg ml⁻¹ for octamers. The ESI-IM-MS technique thus allows detecting distinct A β assemblies, but with a sensitivity that decreases by 2- to 3-fold with the rise of the oligomeric state compared to A β monomers. This has been introduced in the Methods and Results sections (pages 7 and 26).

5. In the *in vivo* experiments in which PrP^{Sc} levels are compared, were all of these measurements made at end stage, regardless of the relative survival times, or at comparable dpi's? Should be specified.

As already mentioned in the figure legends, PrP^{Sc} levels were measured in C57Bl/6J mice at the end stage of prion diseases (Fig. 3C). With prion-infected APP23 mice, PrP^{Sc} levels were measured at the end stage or when sacrificed at 260 dpi (Fig. 6B). This is now specified in the body text.

6. PET analyses, especially as portrayed in Fig 6A: These analyses and data presentation were difficult to understand for the uninitiated. Please provide more explanation of what the data points represent and how the threshold of positivity was determined. An indication of signals from negative controls, and their variability needs to be provided.

As indicated in the legend of Fig. 6A, each data point is one mouse in one animal group. For example, red triangles refer to the APP23 mouse group (n=10) injected with inoculums derived from Fk-1C11 cells supplemented with A β trimers. Top panel of Fig. 6A is a longitudinal follow up (between 2 and 8 mpi) of each PIB-positive mouse. This permits to show the intensity of A β deposition over time for each PIB-positive mouse. Lower panel of Fig. 6A shows survival curves of prion-infected APP23 mice. For one curve, each symbol refers to one dead mouse in one animal group.

To facilitate the understanding of Fig. 6A, we have now introduced 2 tables as supplementary data that show:

- (i) The number of animals with A β deposition for each group (Supplementary Table 1). This table includes negative controls, that is, APP23 mice inoculated with uninfected 1C11

- cells-derived homogenates and APP23 mice inoculated with uninfected 1C11 cells-derived homogenates supplemented with A β trimers.
- (ii) The impact of PDK1 inhibition with BX912 on the number of animals positive for A β deposition (Supplementary Table 2).

The threshold of PET positivity was considered as the limit of quantitation (LOQ) of PIB binding determined by the statistical method developed by Armbruster DA, et al. (1994) [1]. Briefly, according to guidelines of the Clinical and Laboratory Standards Institute (CLSI EP17-A2) [2], PIB bindings were performed on four “control” mice (two females and two males, three months-old, free of A β deposits as assessed by histology), three replicates a week for five weeks with two different lots of PIB, *i.e.*, 120 replicates in total. The limit of detection (LOD) calculated by the adjusted Currie method [3] resulted in 0.05 ± 0.01 % (SD). In the statistical method of LOQ determination, LOQ is 10 SD above LOD, resulting that PIB uptake was considered significant for values superior to 0.15% of the injected dose (ID) per g of animal.

[1] Armbruster DA, Tillman MD, Hubbs LM (1994) Limit of detection (LOD)/limit of quantitation (LOQ): comparison of the empirical and the statistical methods exemplified with GC-MS assays of abused drugs. *Clin Chem* 40, 1233-1238).

[2] CLSI EP17-A2 protocol, evaluation of detection capability for clinical laboratory measurement procedures: approved guideline-second edition (2012) vol 32, N° 8, Wayne, Pennsylvania.

[3] Gibbons RD, Coleman DE, Coleman DD (2001) Statistical methods for detection and quantification of environmental contamination. John Wiley & Sons, chapter 5, pp 48-50.

This has been introduced in the Method section (page 28).

For the negative controls, that is, APP23 mice injected with uninfected 1C11 cells-derived inoculums (n=10), all mice displayed a PIB score < 0.1% of the Injected Dose per g of animal till the end of the experiment (260 days). With APP23 mice injected with uninfected 1C11 cells-derived inoculums supplemented with synthetic A β trimers (n=10), 8 over 10 mice exhibited a PIB score < 0.1% and the two others a PIB score under the limit of positivity (*i.e.*, 0.14% and 0.15%) at day 260.

7. Also, following on the previous concern, additional types of measurements of Abeta deposition (such as neurohistological analyses) should be provided to augment the PET results.

We performed histological experiments now showing A β deposition in the central nervous system of APP23 mice injected with Fk-1C11 cells-derived inoculums supplemented with synthetic A β trimers that is more substantial than in prion-APP23 mice injected with Fk-inoculum without addition of A β trimers. This is now introduced in new Supplementary Figure 2A. Moreover, quantification of A β deposition in the brain of prion-infected APP23 mice revealed that A β deposition occurred mainly in the cortex, and to a lesser extent in the thalamus, the hippocampus, the striatum, and the cerebellum (now introduced in new Supplementary Figure 2B).

8. L333-336: This conclusion is not convincing. It would seem that too few mice getting sick in either group to make much of a statement like this.

Referee 1 refers to the sentence “Similarly, enrichment of APP^{null}-Fk-1C11 inoculums with A β trimers provoked death of prion-infected APP23 mice at early time (>247 dpi) compared to mice infected with APP^{null}-Fk-1C11 inoculums devoid of A β seeds (>260 dpi) (Fig. 6A)”. As requested, this sentence was tone-down and replaced by “Enrichment of APP^{null}-Fk-1C11 inocula with A β trimers provoked the death of two mice out of ten (Fig. 6A, close squares; [newly introduced] Supplementary Table 1), while none of the mice injected with prion inocula free of A β seeds died before they were sacrificed

at 8 mpi (Fig. 6a, open squares; [newly introduced] Supplementary Table 1). This suggests that brain A β deposition induced by A β trimers present in the inocula tends to accelerate the death of prion-infected mice.” (page 16).

9. L336-339: Following on point #5 above, were these measurements all made at the clinical end point? It seems to me that to better support this conclusion, one would need to compare animals at same time point, e.g. at ~180 dpi for +/- trimers with Fk-1C11 inoculum.

The *in vivo* experiments with APP23 mice were designed to delineate whether A β accumulated upon prion infection can deposit in the brain and impacts on mice survival. Assuming that A β trimers co-injected with PrP^{Sc} recruit PrP^{Sc}-induced A β and promote A β deposition, APP23 mice were inoculated with homogenates derived from APP^{null}-Fk-1C11 cells (without A β), and Fk-1C11 cells (with endogenous murine A β) supplemented or not with human A β trimers. We monitored an accelerated death of prion-infected APP23 mice that correlated with the presence of high amounts of A β deposits in the brain (*i.e.* mice inoculated with Fk-1C11 cells-derived homogenates -with endogenous A β - supplemented with A β trimers, Fig. 6A). As the levels of PrP^{Sc} measured in the brain of dead mice or in mice sacrificed at 260 dpi were not significantly different between groups (Fig. 6B), reduced survival of prion-infected APP23 mice was not associated with over-accumulation of PrP^{Sc}. This notion has been introduced in the results section (page 16).

We further excluded a potential effect of inoculated A β on the kinetics of PrP^{Sc} accumulation. In Fig. 5, 1C11^{5-HT} cells infected with prion-inoculums A β -free or not, supplemented or not with A β trimers, exhibited the same level of PrP^{Sc} when all analyzed after 10 days.

Reviewer #2 (Remarks to the Author):

Amyloid plaques containing the Alzheimer's disease-related A β peptide can be found in human prion diseases such as Creutzfeldt-Jakob disease (CJD). However, its relevance to prion disease progression and pathogenesis is unknown. The manuscript by Ezpeleta et al. uses a prion-infected cell line that does or does not express the Alzheimer's precursor protein (APP) to study the possible connection between prion pathogenesis and A β deposition. They show that APP processing by specific proteases is altered during prion infection, in vitro and in vivo, leading to increased production of A β monomers and trimers. Using a transgenic mouse model for A β deposition, they further show that the presence of A β trimers in the prion inoculum leads to A β deposition and more rapid prion disease. They conclude that alterations in A β processing by proteases leads to increased levels of A β deposition which can accelerate prion disease.

While the paper can be somewhat difficult to follow, the data are clearly presented and support the conclusion that PDK1 dysregulation during prion infection can lead to increased levels of A β in tissue culture cells as well as in a transgenic mouse line used to model certain aspects of Alzheimer's disease. However, the manuscript suffers somewhat from overinterpretation of the data. For example, they state that PrP^{Sc} induces overactivation of PDK1 and is sufficient to promote elevated levels of A β in cerebral spinal fluid (lines 228-230 and 200-202). However, only complex prion homogenates are being tested. There are no data showing that PrP^{Sc} is directly mediating either of these effects. To draw this conclusion, experiments utilizing purified PrP^{Sc} would have to be done.

To our knowledge, purifying to homogeneity PrP^{Sc} from prion-infected cells or brain has never been achieved because of complex interactions between PrP^{Sc}, extracellular matrix elements, nucleic acids and proteins, and because of the aggregative properties of PrP^{Sc}. Most people in the prion field work with cell- or brain-derived prion homogenates to infect diverse experimental paradigms, considering that neurotoxic events are caused by PrP^{Sc}. Nevertheless, to avoid some overstatements, "PrP^{Sc}" was replaced by "prion infection" when justified in the manuscript.

The authors need to be more accurate in how they state their conclusions so that they match the data presented.

This has been checked throughout the manuscript. Supplementary Tables 1 and 2 as well as Supplementary Figures 1 and 2 have been introduced and Figs. 1A-D and Fig. 6C have been improved to support the conclusions.

1) In Figure 1E and F, what percentage of the total A β in the cell supernatant of prion infected cells is oligomeric, i.e. trimers or tetramers? In the cell lysate, only 0.2% of the A β was oligomeric (line 136). Is a similarly low percentage of A β oligomeric in the supernatant?

In the culture medium of prion-infected 1C11^{5-HT} neuronal cells, ~2.75% of total A β are multimers with A β trimers representing ~2.5% of total A β and A β tetramers representing ~0.25% of total A β . This is ~10 times more important than in cell lysates. As requested, the percentages of A β trimers and tetramers present in the culture medium of prion-infected 1C11^{5-HT} cells have now been introduced in the results section (page 7).

2) Why weren't uninfected 1C11 cells treated with BX912 and TAPI-2? Wouldn't this be a control for how PDK1 and TACE react to their respective inhibitors in the absence of prion infection?

New Figs. 1A-D now indicate the level of A β 40, A β 42, sAPP α and sAPP β in the culture medium of uninfected 1C11^{5-HT} neuronal cells exposed to the PDK1 inhibitor BX912 (1 μ M) or a combination of BX912 (1 μ M) and the TACE inhibitor, TAPI-2 (100 μ M). The levels of A β 40/42, sAPP α / β were quite

insensitive to these inhibitors in uninfected neuronal cells. PDK1 activity is low in uninfected cells and does not interfere with the trafficking and activity of TACE α -secretase, as already documented in Pietri *et al.* (Nat. Med. 2013; Ref 23). Basal α -cleavage of APP seems to depend on secretases distinct from TACE as also previously observed with plasma membrane TNF α receptors (Pietri *et al.*, Nat. Med, 2013).

3) No methods are given for how the oligomeric state of A β was determined using ion mobility and ESI-IM-MS (lines 125-126). How did they control for the possibility that oligomerization of A β occurred during and/or after preparation of the sample as opposed to within the prion-infected cell or brain?

We now specified in the Method section (p24-26) how the oligomeric state of A β was determined by ESI-IM-MS: "Detection and quantification of A β multimers by ESI-IM-MS was performed as described in ref 27. Approximately 50 μ l of cell culture supernatant or CSF were added to 950 μ l of methanol and acetic acid (1%) solution maintained at 37°C for 30 minutes and then centrifuged for 1 hour at 1500 g. The supernatant was subjected to analysis by ESI-IM-MS without further sample preparation. Electrospray Ionization Mass Spectrometry (ESI-MS) coupled with Traveling Wave Ion Mobility Spectrometry (TWIMS) experiments were performed with a Synapt G2 HDMS mass spectrometer (Waters) equipped with a Z-spray ESI source. Capillary voltage of 2.00 kV, sampling cone voltage of 50 V, extraction cone voltage of 4.0 V, source temperature of 100 °C, desolvation temperature of 200 °C, trap and transfer collision energy of 10 eV were used as parameters for ESI. Although no signal was detected over 3000 m/z, data were acquired over the m/z range of 500 to 5000 for 2 min. Helium flow rate of 30 ml min⁻¹ and nitrogen flow rate of 25 ml min⁻¹ were used for TWIMS experiments, with 250 m s⁻¹ for wave velocity and 8.0 V for wave height. Calibration of the experimental arrival time to determine collision cross section (Ω D) values was performed following previously reported method (Ruotolo BT et al (2008) Nat. Protoc. 3, 1139–1152; Bush MF et al (2010) Anal. Chem. 82, 9557–9565) using ubiquitin and cytochrome c as calibrants. The raw data were processed using Mass Lynx v4.1 software (Waters). The reported data are the average of three independent experiments. The ESI-IM-MS data were assigned to specific A β oligomers on the following basis: (i) First, ion mobility is often described as proportional to collision cross section-to-charge ratio (Ω/z). Any charge state (m/z) in the mass spectrum could correspond to several species (e.g. M+z, D+2z, Tr+3z, Te+4z; with M = monomers, D = dimers, Tr = trimers, and Te = tetramers). These species have the same m/z value and are therefore indistinguishable in the mass spectrum. They can be distinguished by IM because their Ω/z differ. For example, the Te charge would be theoretically four times that of M; however, the Ω Te would be smaller than four times the Ω M because of the favorable interactions between the monomers and higher order species. Thus Ω Te/4z < Ω Tr/3z < Ω D/2z < Ω M/z, *i.e.* the species that have smallest drift times were initially assigned to those corresponding to the largest oligomer. For the A β 40 spectra at m/z 2165, four contributions were detected in the drift time domain at 4.1, 5.1, 7.7, and 12.8 ms. The peaks at 4.1 and 12.8 ms were initially assigned to the highest and lowest oligomer order, respectively. (ii) Second, we assigned a given oligomer order on the basis of the ¹³C isotope distribution associated with each of the peaks separated in the mobility dimension as described in (Kloniecki M et al (2011) J Mol Biol 407,110-124). Applying this consideration to the m/z peak of A β 40 observed at m/z 2165, the ¹³C isotope distribution associated with the mobility peaks detected at 12.8 and 7.7 ms were consistent with charges +2 and +4, respectively, and therefore were assigned to M+2 and D+4, respectively. The resolution of the ¹³C isotope distribution for the mobility peaks observed at 5.1 and 4.1 ms was not sufficient for their assignation. In this case, the charge envelope in the 2D ESI-IM-MS spectrum was analyzed. In the ESI-IM-MS spectra, we found other charge states consistent with the assignment of the peaks at 5.1 and 4.1 ms to Tr+6 and Te+8, respectively. For the former, these were Tr+ 5 and Tr+7

and for the latter Te+7. (iii) Third, the conformation of the ions was also taken into account. Peaks that had the same m/z and the same ^{13}C isotope distribution were attributed to compact and extended forms of the same oligomer. The m/z peak of A β 40 observed at m/z 1732 showed two contributions in the drift time domain (at 4.4 and 7.4 ms), and the ^{13}C isotope distribution associated with these two mobility peaks was consistent with +5 charges, so they were assigned to compact and extended forms of the D+5, respectively.”

In Fig. 1, we measured levels of A β 40 and A β 42 in the cell culture medium of uninfected serotonergic 1C11^{5-HT} cells (Figs. 1A-B), but we never detected A β 40/42 trimers and tetramers (Figs. 1E-F). With prion-infected serotonergic neurons (Fk-1C11^{5-HT}), we monitored a 2- to 4-fold rise of A β 40/42 monomers (Figs. 1A-B) and detected A β 40/42 trimers and tetramers (Figs. 1E-F) in the culture medium. Upon PDK1 inhibition with BX912, A β 40/42 monomers in the culture medium of prion-infected neurons decreased to a level that was quite comparable to that measured at basal conditions with uninfected cells (Fig. 1A-B). Although the amount of A β 40/42 multimers dropped down upon PDK1 inhibition, A β 40/42 trimers, but not tetramers, were still present and measurable in the culture medium of prion-infected cells (Figs. 1E-F). As the levels of A β trimers and tetramers depend on the upstream amount of A β monomers, our data strongly suggest that A β multimers are not produced upon sample preparation, but are clearly associated with prion infection. To our knowledge, there is no tool to probe for the presence of A β multimers *in situ* before their extraction.

4) How do the authors explain the somewhat paradoxical result that, when compared with prion-infected cells that did express APP, much higher levels of A β tetramers were found in mice inoculated with prion-infected cells that did not express APP? It’s not entirely clear from the results how they are interpreting this result.

This point has not been solved in our study and remains intriguing. With prion-infected inocula free of A β (APP^{null}-Fk), A β 40/42 peptides generated in prion-infected C57Bl6/J mice seem to assemble primarily and spontaneously into tetramers. When A β (including A β trimers) is present in prion-infected inocula (Fk), then A β 40/42 peptides generated upon prion infection assemble mostly into trimers. This suggests that A β seeds in the inocula (likely the A β trimer) would drive the assembly of A β into trimers. As A β trimers would potentially display “prion-like” properties, inoculated A β trimers would provoke the assembly of A β monomers into trimers following the law of action mass, and thereby self-propagate. As this was too speculative, we decided not to introduce these concepts in our manuscript.

5) If available, it would be informative to have either the PET scans for the data in Figure 6A or histopathological confirmation of A β deposition in the APP23 mice infected with prions. While the data in Figure 6A give a quantitative measurement of A β , it would be of interest to know the regions of the brain where A β has deposited as amyloid.

As also requested by referee 1, we now introduced Supplementary Figure 2 showing quantification of the β -amyloid load in the brain of prion-infected mice. This quantification revealed that A β deposition occurred mainly in the cortex, and to a lesser extent in the thalamus, the hippocampus, the striatum, and the cerebellum (Supplementary Figure 2B). This likely reflects the cortico-tropism of the Fukuoka-1 prion strain (Arima, K et al. J Virol. 2005 [1]) and thereby the strong incidence of prion infection on the production of A β in this brain area. These supplementary data also confirm that the A β deposition in the brain of APP23 mice injected with Fk-1C11 cells-derived inoculums supplemented with synthetic trimers of A β was more substantial than in the brain of prion-infected

APP23 mice that did not receive A β trimers (Supplementary Figure 2A). This is now indicated in the results section (page 15).

[1] Arima K, Nishida N, Sakaguchi S, Shigematsu K, Atarashi R, Yamaguchi N, Yoshikawa D, Yoon J, Watanabe K, Kobayashi N, Mouillet-Richard S, Lehmann S, Katamine S. *Biological and biochemical characteristics of prion strains conserved in persistently infected cell cultures. J Virol.* 2005 Jun;79(11):7104-12.

6) In line 250, the term “PrPres” is used. Does this refer to the PrP^{Sc} data in Figure 5A? If so, the terms should match.

The term PrP^{res} refers to PrP^{Sc} that resists to digestion by proteinase K. The proteinase K digestion assay permits to prove the presence of PrP^{Sc} in any samples (see Methods section). Nevertheless, and to avoid confusion, the term PrP^{res} was removed and replaced by “PrP^{Sc}” to match with the results displayed on Figs 5A.

Reviewer #3 (Remarks to the Author):

The authors investigate functional aspects of the observed Abeta histopathology in infectious prion disease. This is especially interesting given the emerging relationship between PrP and Abeta in Alzheimer's disease (AD). Specifically, the authors examine how Abeta peptides derived from prion-infected cells affect prion infectivity (they don't), how prion infection affects Abeta production (PDK1-TACE dependently), and how Abeta influences prion-dependent mortality (accelerates it).

Overall the study feels incomplete in each of the areas of investigation. And while the role of Abeta in prion disease is an interesting and important topic, the insights here are incremental and somewhat fragmentary in their demonstration.

The oligomeric state of the Abeta in this study is poorly characterized. This is important because the predominant PrP-interacting species of Abeta oligomer in both human AD brain and mouse AD model brain are high molecular weight aggregates in the 100-mer range, observable via size exclusion chromatography or native PAGE. The authors declare no higher order oligomers of Abeta (above tetramer) were evidenced in samples, but show no data to support that claim. On what basis is this conclusion reached? The presence of Abeta trimers and tetramers in the absence of larger oligomers would be notable and requires thorough demonstration through more than ESI-IM-MS. In general it would be important to show the overall similarity of all oligomeric species for brain extracts, CSF, and cell-based inocula.

According to Pujol-Pina *et al.* (Scientific reports, 2015), the ESI-IM-MS method emerges today as the most sensitive approach to detect and quantify A β 40/42 oligomers ranging from dimers to hexamers in a preparation of pure synthetic human A β 40 and A β 42. Pujol-Pina *et al.* also claimed that the ESI-IM-MS method does not generate "artifacts", that is, A β assemblies that do not exist in biological samples, on the contrary to the SDS-PAGE analysis. See also our answer to point 3 raised by referee 2.

When calibrating the ESI-IM-MS method, we found that the limits of detection (LOD) of A β monomers and oligomers by the ESI-IM-MS technique were 0.09 pg ml⁻¹ for A β 40/42 monomers, 0.10 pg ml⁻¹ for dimers, 0.12 pg ml⁻¹ for trimers, 0.16 pg ml⁻¹ for tetramers, 0.20 pg ml⁻¹ for pentamers, 0.25 pg ml⁻¹ for hexamers, 0.28 pg ml⁻¹ for heptamers and 0.29 pg ml⁻¹ for octamers (now specified in the Methods section). The ESI-IM-MS technique thus allows detecting distinct A β assemblies, but with a sensitivity that decreases by 2- to 3-fold with the rise of the oligomeric state compared to A β monomers. Keeping in mind such differences in LODs between A β multimers, we concluded for the absence (or below the LODs) of soluble A β oligomers with an oligomeric state higher than tetramer in lysates of prion-infected cells (Fig. 2A), their culture medium (Figs. 1E-F, Figs. 2A-B) or in the CSF of prion-infected C57Bl/6J mice (Figs. 4E-F). In addition, our data clearly show the presence of A β trimers/tetramers in quite similar proportions in both the cell supernatant of Fk-infected 1C11^{5-HT} cells and the CSF of C57Bl/6J mice injected with Fk inoculums (Figs. 1E-F, Figs. 4E-F).

The study does not address how Abeta exerts its toxic action. Is it via PrP^C-binding as in AD? This should be explored.

Our data show accelerated death of prion-infected APP23 mice notably when A β trimers were co-inoculated with PrP^{Sc} (Fig. 6A).

In an Alzheimer's context, we previously showed in transgenic mice with AD-like pathology (the Tg2576 model) that the increased brain A β levels and thereby the cognitive and memory deficits depended on PrP^C signaling and the rise in PDK1 activity (Pietri *et al.*, Nat. Med. 2013). siRNA-

mediated silencing of PrP^C in “AD neurons” (i) decreased PDK1 activity back to basal level (as in non-AD neurons), (ii) rescued TACE APP α -processing, and (iii) counteracted memory and cognitive impairments in those mice (Pietri *et al.*, Nat. Med., 2013).

Here we show that outside the context of prion infection synthetic A β trimers fail to induce any augmentation of PDK1 activity in APP23 mice injected with those A β trimers (data now introduced in Fig. 6C), suggesting that A β trimers do not mobilize PrP^C and do not recruit PrP^C signaling. This is further supported by the absence of A β accumulation in 1C11^{5-HT} serotonergic neurons exposed to synthetic human A β 40/42 trimers (Figs. 5B-C). As rightly outlined by referee 3, and in agreement with several studies (Lauren, 2009; Um, 2012; Kostylev, 2015), only A β oligomers with high molecular weight in the 20-mer to 100-mer range were shown to bind PrP^C and to provoke neuronal dysfunctions through PrP^C in Alzheimer’s disease.

Investigating whether PrP^C would mediate A β toxicity within a prion-infectious context cannot be accomplished as PrP^C silencing or knock out will impair prion infection and thereby all downstream events (including A β production) induced by PrP^{Sc}. (Please see also the answer to the penultimate concern you raised).

In figure 1, it is difficult to interpret the relationship between TACE and PDK1 because there is no condition where the TACE TAPI-2 is applied alone to FK-1C11 cells. It would also be informative to show here whether TAPI-2 and BX912 have an effect on 1C11 cells in the absence of prion infection. Is the role of TACE and PDK1 completely prion dependent?

Outside of a prion infectious context, we previously showed using plasma membrane TNF α receptors (TNFRs) as readout that basal α -cleavage of TNFRs did not depend on PDK1 activity or TACE α -secretase activity (see Pietri *et al.*, Nat. Med, 2013).

All the controls requested by referee 3 were now introduced in Figs. 1A-D. These data show that within an uninfected context (i) basal PDK1 activity is not at play in APP processing and (ii) secretases distinct from TACE assume basal APP α -cleavage. See also the answer to referee 2, point 2.

The unconventional use of asterisks denoting statistical significance is confusing. The use of one asterisk (*) to denote p<0.001 with two asterisks (***) to denote p<0.01, is inverted from convention.

The asterisks were changed to match with their conventional use in statistical significance tests.

The conclusion that the beta secretase pathway is being favored by PrP^{Sc} seems likely but needs to be directly demonstrated with manipulation of beta secretase genetically or pharmacologically.

Prion-infected 1C11^{5-HT} neuronal cells (Fk-1C11^{5-HT}) were treated with the BACE1 pharmacological inhibitor MK-8931 (1 μ M) for 1 h. New Supplementary Figure 1 now shows that inhibiting BACE1 reduces the level of A β 40, and A β 42, and sAPP β in the culture medium of prion-infected neurons. On the opposite, we monitored an increase in sAPP α level in the culture medium of prion-infected cells treated with MK-8931. Despite strong reduction (more than 80%) of plasma membrane TACE level in prion-infected cells (Pietri *et al.*, Nat. Med. 2013), the increase in sAPP α level that typically accompanies BACE1 inhibition likely results from augmented bioavailability of APP for residual α -secretase activity, as previously observed by Villarreal *et al.* in mouse model with Alzheimer-like pathology (Tg2576) treated with MK-8931 (2017). These data have been introduced page 6 of the manuscript.

Villarreal S, Chronic Verubecestat Treatment Suppresses Amyloid Accumulation in Advanced Aged Tg2576-AβPPswe Mice Without Inducing Microhemorrhage. Journal of Alzheimer's disease 59 (2017): 1393-1413.

The assertion that Abeta trimers, likely acting as seeds, can boost the formation of Abeta multimers within a prion infectious context implies that the prion infection enables the action of Abeta trimers. This would be better supported if the action of Abeta trimers were examined outside the context of prion infection. Are they able to boost multimers in the absence of prion infection or is there interaction between the trimers and prion? AD model mice can have accelerated mortality. Are trimers able to accelerate death independently of prion infection, and thus suggest additive parallel mortality-enhancing pathways? Or does the mortality enhancement of trimers depend on prion co-infection, suggesting mechanistic convergence? Without examining the effect of Abeta independently, it is difficult to draw hard conclusions.

The action of Aβ trimers was already examined outside the context of prion infection in our previous version of the manuscript (pages 11-13). In Fig. 5, we showed that incubation of uninfected 1C11^{5-HT} serotonergic neuronal cells with synthetic human Aβ40/42 trimers for ten days had no impact on the basal production of Aβ40 and Aβ42 monomers (Figs. 5B-C). We also monitored that Aβ trimers did not induce the production of Aβ multimers by uninfected neuronal cells (Fig. 5D). After cell exposure to Aβ trimers for 10 days, Aβ multimers found in the culture medium of uninfected 1C11^{5-HT} cells corresponded to inoculated Aβ trimers, the levels of which were lower than those introduced, suggesting partial degradation of Aβ trimers.

As mentioned above, Aβ trimers do not corrupt the PrP^C-PDK1 signaling axis in APP23 mice injected with uninfected inocula supplemented with synthetic Aβ trimers (new data introduced Fig. 6C).

As also requested by referee 1, the newly introduced supplementary table 1 shows that none of the APP23 mice injected with prion-free inoculums (1C11) supplemented with synthetic Aβ trimers were positive for Aβ deposition nor died from 2 to 8 mpi.

As rightly outlined by referee 3, the mortality enhancement of synthetic Aβ trimers depends on prion co-infection. We found that the anticipated death of prion-infected mice correlated with an increased intensity/velocity of brain Aβ deposition (Fig. 6A). Of note, prion infection alone (APP^{null}-Fk inoculums) while promoting a rise of Aβ40/42 through deregulation of the PDK1-TACE pathway (Figs. 4A-B) is not sufficient to provoke Aβ deposition. Aβ deposition depends on the presence of Aβ seeds (including Aβ trimers) in the inocula. We thus proposed a two-step model where (i) PrP^{Sc} first stimulates the production of seedable Aβ that (ii) precedes Aβ deposition driven by Aβ seeds. This notion was already introduced in our manuscript in the discussion section (pages 20-21).

It would be advisable for the authors to enlist the services of an editor with English as a first language. In places, grammatical peculiarities make interpretation challenging.

Native English-speaking people read the manuscript. This new version of our manuscript takes into account all their grammar corrections.

REVIEWERS' COMMENTS:

Reviewer #1 (Remarks to the Author):

The authors have addressed my concerns adequately.

Reviewer #2 (Remarks to the Author):

None.

Reviewer #3 (Remarks to the Author):

The manuscript has been edited with attention to clearer English which makes the work more easily interpretable.

There are some fundamental matters requiring resolution.

The authors have incompletely characterized the Abeta of the study, both in the inoculum used and the Abeta generated in vivo. This deficiency must be addressed for the meaning of the results to be understood in the context of the current amyloid field. There are several aspects in need of attention:

1. The multimeric state of the Abeta species in the supernatant of Fk-1C11 cells are described, while structurally uncharacterized cell lysates are used for inoculation. For the results to be understood, the same material should be used for both Abeta molecular weight characterization and inoculation.
2. As stated in my earlier comments, higher order oligomers must be characterized for all Abeta analyses. The authors responded that SDS PAGE produces artefactual aggregates, which is true and why I suggested using non-denaturing native PAGE. The authors are aware of methods to characterize high MW Abeta because they cite several papers that do so, in response to my question about the mechanism of toxicity. Fully characterizing the Abeta species in this study is not a trivial detail, especially in light of the difference they observe in survival of mice inoculated with Fk-1C11 or APPnull-Fk-1C11 + Abeta trimers in Fig 6A. Differences in Abeta MW species between cell-generated Abeta and synthetic preps could account for the disparity. Alternatively, there may be a difference in infectious PrPsc species when forming in the presence of Abeta, potentially complexing with and incorporating Abeta since PrPc is well known to bind Abeta. This, also, should be examined in their lysates.

Regarding the role of PrP in the mechanism of toxicity, the authors state that it cannot be investigated in this experimental context because PrP knock out would impair infectivity. The authors should employ other methods besides knock out to investigate mechanistic differences between inoculate effects in vivo. The references the authors cited are a good place to start in identifying candidate mechanisms and biochemical methods.

In Fig 6B, is the difference in brain PrPsc amount between mice treated with APPnull-Fk-1C11 and mice treated with APPnull-Fk-1C11 + Abeta trimers statistically significant? This should be indicated.

REBUTTAL LETTER to REVIEWER'S COMMENTS

Reviewer #3

The authors have incompletely characterized the A β of the study, both in the inoculum used and the A β generated *in vivo*. This deficiency must be addressed for the meaning of the results to be understood in the context of the current amyloid field. There are several aspects in need of attention:

1. The multimeric state of the A β species in the supernatant of Fk-1C11 cells are described, while structurally uncharacterized cell lysates are used for inoculation. For the results to be understood, the same material should be used for both A β molecular weight characterization and inoculation.
2. As stated in my earlier comments, higher order oligomers must be characterized for all A β analyses. The authors responded that SDS PAGE produces artefactual aggregates, which is true and why I suggested using non-denaturing native PAGE. The authors are aware of methods to characterize high MW A β because they cite several papers that do so, in response to my question about the mechanism of toxicity. Fully characterizing the A β species in this study in not a trivial detail, especially in light of the difference they observe in survival of mice inoculated with Fk-1C11 or APP^{null}-Fk-1C11 +A β trimers in Fig 6A. Differences in A β MW species between cell-generated A β and synthetic preps could account for the disparity. Alternatively, there may be a difference in infectious PrP^{sc} species when forming in the presence of A β , potentially complexing with and incorporating A β since PrP^{sc} is well known to bind A β . This, also, should be examined in their lysates.

A β generated *in vivo* was already characterized in our previous versions of the manuscript. As indicated pages 9 to 10 of the result section, monomers of A β 40 (7500 ± 225 pg ml⁻¹) and A β 42 (900 ± 32 pg ml⁻¹) were mainly detected in the CSF of C57Bl/6J mice inoculated with Fk-1C11 cells (Figs 4A-B). ESI-IM-MS analyses also revealed the presence of A β 40 and A β 42 trimers (150.0 ± 3.7 and 33.0 ± 3.4 pg ml⁻¹ respectively) and tetramers (11.2 ± 2.2 and 11.5 ± 1.3 pg ml⁻¹ respectively) (Figs 4E-F) that together represent 0.20% over total A β species.

As requested, lysates of Fk-1C11 cells used for mouse inoculation were characterized for their content in A β . Fig. 2A already showed histogram quantifications of monomers of A β 40 and A β 42 in Fk-1C11 cell lysates. New ESI-IM-MS analyses reveal the presence of a mix of trimers and tetramers of A β 40 and A β 42 in those lysates. Note that because of the complexity of the sample matrix, and the detection of trimers and tetramers of A β 40/42 out of the linearity zone of the ESI-IM-MS approach, accurate quantifications of A β multimers in Fk-1C11 lysates could not be done. The characterization of A β multimers in Fk-1C11 cell lysates by non-denaturing native PAGE would necessitate to concentrate the samples, a step that would likely change the equilibriums between A β species and thereby bias the analysis.

As compared to the Fk-1C11 inoculum, we agree that supplementation of APP^{null}-Fk-1C11 inoculum with synthetic A β trimers is a partial complementation of the inoculum. Even if there is disparity regarding the kinetics and intensity of A β deposition, and the survival of APP23 mice injected with Fk-1C11 inoculum vs. APP^{null}-Fk-1C11 inoculum

enriched with A β trimers, our data strongly support the view that A β trimers co-injected with PrP^{Sc} play a role in prion pathogenesis. It is however unlikely that the disparity originates from difference in infectious PrP^{Sc} species, due to potential interaction between PrP^{Sc} and A β , as we showed no difference in motor deficit and survival of C57Bl/6J mice injected with inocula derived from Fk-1C11 cells (containing A β) or APP^{null}-Fk-1C11 cells (without A β). To our knowledge, a direct physical interaction between PrP^{Sc} and A β species has never been evidenced *in vivo* and remains an open question.

Regarding the role of PrP in the mechanism of toxicity, the authors state that it cannot be investigated in this experimental context because PrP knock out would impair infectivity. The authors should employ other methods besides knock out to investigate mechanistic differences between inoculate effects *in vivo*. The references the authors cited are a good place to start in identifying candidate mechanisms and biochemical methods.

This point has already been addressed in our previous rebuttal letter (points 2 and 6). Although interesting, we deem that this specific request falls beyond the scope of the present study.

In Fig 6B, is the difference in brain PrP^{Sc} amount between mice treated with APP^{null}-Fk-1C11 and mice treated with APP^{null}-Fk-1C11 + Abeta trimers statistically significant? This should be indicated.

To improve clarity, the sentence "brain levels of PrP^{Sc} were highly comparable different between each prion-infected mice group at the end stage of the disease or when mice were sacrificed at 8 mpi (Fig. 6B)" was changed by "brain levels of PrP^{Sc} were not statistically different between each prion-infected mice group at the end stage of the disease or when mice were sacrificed at 8 mpi (Fig. 6B)" (see results section page 15).